# Investigation of non-equilibrium turbulence decay in the atmospheric boundary layer using Doppler lidar measurements

Maciej Karasewicz[1], Marta Wacławczyk[1], Pablo Ortiz-Amezcua[1], Łucja Janicka[1], Patryk Poczta[1,2], Camilla Borges Kassar[1], and Iwona S. Stachlewska[1]

[1]Institute of Geophysics, Faculty of Physics, University of Warsaw, Pasteura 5, Warsaw, Poland
[2]Laboratory of Bioclimatology, Faculty of Environmental and Mechanical Engineering, Poznan University of Life Sciences, Piątkowska 94, Poznan, Poland

**Correspondence:** Marta Wacławczyk (marta.waclawczyk@fuw.edu.pl)

**Abstract.** This work concerns analysis of turbulence in the Atmospheric Boundary Layer (ABL) shortly before and after the sunset. Based on a large set of the Doppler lidar measurements at rural and urban sites we analyze frequency spectra of vertical wind at different heights and show that they increasingly deviate from the $-5/3$ Kolmogorov's prediction in the measured low-wavenumber part of the inertial range. We find that before the sunset the integral length scales tend to decrease with time. These

findings contrast with a classical model of equilibrium decay of isotropic turbulence, which predicts that the scaling exponent should remain constant and equal to $-5/3$ and the integral length scale should increase in time. We explain the observations using recent theories of non-equilibrium turbulence. The presence of non-equilibrium suggests that classical parametrization schemes fail to predict turbulence statistics shortly before the sunset. By comparing the classical and the non-equilibrium models we conclude that the former may underestimate the dissipation rate of turbulence kinetic energy in the initial stages of

decay.

## 1    Introduction

Turbulence in the atmospheric boundary layer (ABL) undergoes temporal changes with the diurnal cycle. After dawn and under clear sky, the surface heating produces convection and a boundary layer starts to grow. Shortly before the sunset, convective ABL collapses rapidly and next a stable nocturnal BL is formed (Nieuwstadt and Brost, 1986).

During day, turbulence production due to buoyancy is prevalent. In the afternoon, the buoyancy flux decreases gradually and eventually becomes negative (Sorbjan, 1997). The time when the heat flux crosses the zero level was identified by Nadeau et al. (2011) as the beginning of the evening transition. At this time instant, turbulence in the ABL starts to decay more rapidly than in the afternoon. After sunset, turbulence is still produced by shear, and remaining thermal forcing, however, mostly in a region close to the surface.

Turbulence in the atmosphere, far enough from the surface, is usually assumed to be approximately homogeneous and isotropic at scales smaller than the integral length scale. In spite of the considerable simplification, homogeneous and isotropic turbulence is a subject of ongoing research, due to its importance for existing theories (Sagaut and Cambon, 2018). Recently, a number of theoretical works addressed the parametrization of decaying isotropic turbulence (Vassilicos, 2015; Goto and

Vassilicos, 2016; Bos and Rubinstein, 2018). In particular, deviations from the predictions of the Kolmogorov theory were observed in the initial stages of decay, after the forcing was switched off. The Kolmogorov's theory of turbulence is of utmost importance as it is widely used to estimate turbulence kinetic energy dissipation rate $\epsilon$ from measured signals. The dissipation rate determines how fast the kinetic energy of turbulence is transferred into heat at the smallest scales, which are of the order of millimeters in the atmospheric turbulence. Hardly ever can such small scales be measured with satisfactory accuracy in the free atmosphere. For this reason, the dissipation rate is estimated indirectly, by assuming that the energy injected at large scales by forcing is transported at a constant rate from larger to smaller eddies. This process is known as the energy cascade. Taking such assumption, Taylor (1935) formulated the famous relation between the dissipation rate $\epsilon = 2\nu \langle s_{ij} s_{ij} \rangle^{1/2}$, where $s_{ij} = (\partial u_i'/\partial x_j + \partial u_j'/\partial x_i)/2$, the turbulence velocity scale $\mathcal{U} = (\langle u_i' u_i' \rangle/3)^{1/2}$, where $u_i'$ is the $i$-th component of fluctuating velocity, and the characteristic length scale of large eddies $\mathcal{L}$ (the integral length scale),

$$\epsilon = C_\epsilon \frac{\mathcal{U}^3}{\mathcal{L}}, \tag{1}$$

where $C_\epsilon$ was believed to be a constant, $C_\epsilon \approx 0.5$. Equation (1) is a basis of many turbulence parametrization schemes.

The validity of the Taylor law was questioned in recent theoretical works and experimental observations, cf. Vassilicos (2015). In particular, it was observed that at the onset of decay, dissipation rate followed a non-standard relation

$$\epsilon = C_{n\epsilon} \mathcal{U}_0 \mathcal{L}_0 \frac{\mathcal{U}^2}{\mathcal{L}^2}, \tag{2}$$

where $C_{n\epsilon}$ is a constant and $\mathcal{U}_0$ and $\mathcal{L}_0$ denote initial values of the turbulence velocity scale and the length scale. Bos and Rubinstein (2018) argued that the appearance of Eq. (2) is connected with deviations from the $-5/3$ scaling of the frequency spectra in the low wavenumber part of the inertial range. They are observed when the energy transfer rate across the scales is not constant, due to a sudden change of a forcing. This state will be further referred to as 'non-equilibrium'. The same notion is used in theromodynamics and fluid mechanics to describe states after a sudden change of external conditions when the system evolves towards another equilibrium (Wacławczyk, 2021). Here, 'equilibrium' is related to the Kolmogorov's turbulence characterized by the $-5/3$ law or $C_\epsilon \approx const$. 'Non-equilibrium', on the other hand, denotes deviations from these laws.

The non-equilibrium scaling was observed in a number of laboratory and numerical experiments (Valente and Vassilicos, 2012; Obligado and Vassilicos, 2019; Steiros, 2022a; Zheng et al., 2023; Obligado et al., 2022; Steiros, 2022a) as well as in the atmospheric turbulence (Wacławczyk et al., 2022). The latter work concerned analysis of airborne measurement data from the stratocumulus-topped boundary layers (STBL). Non-equilibrium dissipation scaling of the form close to (2) was observed, especially near the surface and inside clouds in the decoupled STBL. As discussed therein, weaker turbulence was characterized by larger values of $C_\epsilon$. In particular, $C_\epsilon$ tended to be larger in decoupled STBL, when turbulence was too weak to mix air over the entire height of BL. This was in line with previous findings by Nowak et al. (2021) who speculated that turbulence in the decoupled STBL might be decaying.

To the best of the authors' knowledge, the concept of non-equilibrium was not discussed in the context of the collapse of the convective BL, although previous studies, when analysed from this new perspective, deliver strong indication of the non-equilibrium turbulence decay. For example, Lothon et al. (2014) analyzed data from the Boundary Layer Late Afternoon

and Sunset Turbulence (BLLAST) field experiment and calculated the integral length scales $\mathcal{L}_w$ from the measured vertical velocity. As reported by those authors, $\mathcal{L}_w$ initially decreased in the surface layer and next sharply increased after 7p.m.. In the mixed layer $\mathcal{L}_w$ at first remained constant and then started to increase around 5p.m. In the classical equilibrium turbulence the integral length scale is expected to increase in time during the decay. In the non-equilibrium decay, one the other hand, it may initially decrease and next increase in time (Steiros, 2022b). This scenario is consistent with results of Lothon et al. (2014) in the surface layer.

Results of BLLAST experiment were analyzed also by Darbieu et al. (2015) and compared to results of numerical simulations. In both cases the authors found deviations from the Kolmogorov $-5/3$rd law before the sunset, although validity of the Taylor law (1) was not discussed therein. El Guernaoui et al. (2019) discussed time evolution of the kinetic energy spectra in numerically simulated ABL. They argued that the decay of the kinetic energy is not uniform across the scales and that the largest scales are the most affected.

Dissipation rates in the surface layer during the afternoon and evening transition were reported by Nilsson et al. (2016). The authors investigated relation similar to (1), but with the dissipation length $l_\epsilon$ instead of the integral length scale $\mathcal{L}$. They concluded that assuming $l_\epsilon \propto z$, where $z$ denoted the height, is not sufficient to parametrize dissipation rate in the surface layer. Instead, they proposed to relate $l_\epsilon$ to both $z$ and the height of the boundary layer. Lampert et al. (2016) studied anisotropy of turbulence during the evening transition and reported that the standard deviation of the vertical velocity decreases with time faster than the corresponding standard deviations of horizontal components. This observation could indicate that the non-equilibrium affects spectra of vertical velocity component more than the horizontal ones.

The current work focuses on the decay of turbulence before the sunset, in order to investigate whether it can be parametrized with the non-equilibrium laws. We analyze data from wind Doppler lidar, which has become a strategic instrument in atmospheric research because it provides vertical profiles of radial wind component with high spatial and temporal resolution. Our aim is to examine deviations from the Kolmogorov scaling during the evening transition, at different heights within the ABL and over different environments. Atmospheric turbulence, which is characterized by huge Reynolds numbers, is an ideal testbed for verifying turbulence theories. On the other hand, the theories aim to improve turbulence parametrization schemes, which is of importance for Numerical Weather Prediction and Climate Models.

It is challenging to provide lidar data with sufficient resolution to recognize deviations from the Kolmogorov's scaling in the low-wavenumber part of the inertial range. The spectra are also affected by the filtering (averaging) in space and low signal-to-noise ratio. Hence, these effects should be carefully differentiated. To examine deviations from the Kolmogorov scaling we also analyze the second order structure functions. They are mathematically equivalent to the one-dimensional spectra, but may respond differently to errors due to finite frequency of measurements and due to spatial averaging (Schröder et al., 2024; Wacławczyk et al., 2017, 2020). In this work we investigate the scaling of both, frequency spectra of vertical wind and the structure functions to asses how they change during the decay of turbulence before the sunset. Moreover, we calculate the standard deviation of the vertical velocity and integral length scale and study how they change in time. We also compare the dissipation rates predicted by the classical (1) and the non-equilibrium relations (2).

The diversified large datasets of sufficient resolution, investigated in this work offer a unique possibility to describe the main differences between decay of turbulences generated over urban and rural environments. The rural environment resembles quasi-ideal laboratory conditions for a turbulent flow. The atmospheric boundary layer over an urban environment can experience more mechanical turbulences generated by the wind shear due to the urban friction and its surface roughness, and more thermally generated turbulences. The latter are caused by the interactions of the surface with solar radiations (Svensson, 2004; Edokpa and Nwagbara, 2018). Of particular importance is the thermal heat capacity of surfaces and the related urban heat island phenomenon, which is an effect of the heat accumulation in and over an urban area (Oke, 1987). As reported by Nadeau et al. (2011), the decay of turbulence kinetic energy scales with the characteristic time of the heat flux decay. This time scale is smaller for surfaces that cool down more rapidly. In our work we compare the results at both sites (rural and urban) to assess how surface heterogeneity and surface heat capacity affect turbulence properties.

The paper is structured as follows. The theory of equilibrium and non-equilibrium decay is presented in Section 2. In Section 3 experimental sites and instrumentation are described. Section 4 is devoted to methodology, this is followed by Section 5 with the data analysis. Finally, conclusions and perspectives are discussed in Section 6.

## 2  Theory

### 2.1  Non-equilibrium spectra and structure functions

The theory of Kolmogorov is the foundation of turbulence research (Pope, 2000). It states that at sufficiently large Reynolds numbers and under the assumption of the local isotropy, there exist a range of scales where statistics of velocity take a self-similar form. Further, within this range, a subrange of scales of size $r$, where $\eta \ll r \ll \mathcal{L}$, can be distinguished. It will be further referred to as the 'inertial range'. In this range of scales statistics of turbulence do not depend on viscosity, but only on the dissipation rate $\epsilon$. It follows that the wavenumber spectrum of the vertical wind velocity component can be expressed as

$$E_\perp(\kappa, t) = C_\perp \kappa^{-5/3} \epsilon^{2/3}, \tag{3}$$

where $C_\perp \approx 0.65$ is a constant and $\kappa$ is the wavenumber. Equivalently, the same can be presented in terms of the second-order structure function, which for the vertical velocity component $w$ reads

$$\langle \delta w^2 \rangle = \langle (w(\mathbf{x} + \mathbf{r}, t) - w(\mathbf{x}, t))^2 \rangle.$$

Within the inertial range and under the assumption of local isotropy this function takes the form

$$\langle \delta w^2 \rangle = C (\epsilon r)^{2/3}, \tag{4}$$

where $C \approx 2.86$. Equations (3) and (4) are a basis of various schemes for the estimation of the energy dissipation rate, also from lidar measurements (O'Connor et al., 2010; Lothon et al., 2009; Sanchez Gomez et al., 2021)

Equations (3) and (4) work well when turbulence is close to isotropic, at least locally and is stationary. Recently, extensions of the Kolmogorov's theory towards unsteady turbulence were put forward by Vassilicos (2015). These extensions predict

that the rate of energy transfer across scales in the inertial range is affected by turbulence decay and is not constant. Bos and Rubinstein (2018) expressed the turbulence kinetic energy spectrum as a sum of the equilibrium, Kolmogorov spectrum and a non-equilibrium correction. They derived a formula similar to Eq. (2) and argued that deviations of the spectra from the Kolmogorov's scaling are related to the deviations from $C_\epsilon = const$ in Eq. (1). Goto and Vassilicos (2016) focused on the large-scale part of the turbulence kinetic energy spectrum during the non-equilibrium decay and found that it has the self-preserving form

$$E(\kappa, t) \propto \epsilon \mathcal{L}^3 f(\kappa \mathcal{L}). \tag{5}$$

Steiros (2022a) introduced the notion of "balanced nonstationary turbulence" where the transfer across the scales was proportional to dissipation, however, with a proportionality constant smaller than 1. Such assumption led to the modified form of the energy spectrum in the inertial range

$$E(\kappa, t) = C_k \epsilon(t)^{2/3} \kappa^{-5/3} \left[ 1 - c(\kappa \mathcal{L}(t))^{-2/3} \right]^2 \tag{6}$$

where $c$ is a dimensionless constant. The spectra followed the above formula even during the equilibrium decay with $C_\epsilon = const$. The function brackets in the above equation reaches the value 1 asymptiotically, at large wavenumbers (small scales), where the spectra remain close to the Kolmogorov $-5/3$ form.

Obligado and Vassilicos (2019) investigated how the inertial range of the structure functions is affected during non-equilibrium decay of turbulence. They concluded that in the case of decaying turbulence, the second and the third order structure functions are closest to the Kolmogorov's predictions at the small-scale end of the inertial range. For larger scales, the structure functions increasingly deviate from equilibrium, even at very large Reynolds numbers. The authors considered the Lundgren's formula for the structure functions, derived with the use of matched asymptotic expansions. For very high Reynolds numbers it reads

$$\langle \delta u^2 \rangle = C (\epsilon r)^{2/3} \left[ 1 - A_2 (r/\mathcal{L})^{2/3} \right], \tag{7}$$

where $A_2$ is a dimensionless constant of the order 1. Under the assumption of local isotropy, the above formula is mathematically equivalent to Eq. (6), provided that the bracketed term in Eq. (6) can be expanded in the Taylor series.

We now discuss how turbulence statistics, in particular, the integral length scale change in time according to the theory of equilibrium and non-equilibrium decay. Based on this we can identify which type of parametrization more adequately describes the collapse of the convective boundary layer before the sunset. We mostly refer here to the recent papers by Goto and Vassilicos (2016) and Steiros (2022b) who investigated decaying turbulence using numerical experiments and derived non-equilibrium decay laws.

## 2.2 Equilibrium decay

Under the assumption of horizontal homogeneity the transport equation for the turbulence kinetic energy $k = \langle u_i' u_i' \rangle / 2 = 3/2 \, \mathcal{U}^2$ in ABL reads (Pope, 2000)

$$\frac{\partial k}{\partial t} = -T - \epsilon + P + B, \tag{8}$$

where

$$T = \frac{1}{\rho}\left(\frac{\partial\langle p'w'\rangle}{\partial z} + \frac{1}{2}\frac{\partial\langle u_i'u_i'w'\rangle}{\partial z}\right), \qquad P = -\langle u_i'w'\rangle\frac{\partial\langle u_i\rangle}{\partial z}, \qquad B = \langle w'b'\rangle$$

stand for the turbulent transport, shear production and the buoyancy forcing, respectively. Above, $p'$ denotes the pressure fluctuations and $b'$ the fluctuations of buoyancy. The buoyancy flux will be defined as

$$\langle w'b'\rangle = \frac{g}{\langle\theta_v\rangle}\langle w'\theta_v'\rangle, \tag{9}$$

where $\theta_v$ is the virtual potential temperature and $g$ stands for the gravity acceleration.

Turbulence production due to shear $P$ is an important part of the budget close to the Earth's surface. Buoyancy $B$ plays a dominant role during daytime convection, finally, the role of turbulent transport $T$ is to transfer the kinetic energy produced close to the surface to higher altitudes.

At the beginning of the evening transition, the buoyancy flux $B = 0$ (Nadeau et al., 2011) and the convective boundary layer collapses rapidly. Turbulence is still produced by the shear $P$ in the surface layer, however we can assume that at larger altitudes turbulence starts to decay freely, such that the time derivative of $k$ on the LHS of Eq. (8) is balanced mainly by the dissipation $\epsilon$. Under such assumptions Eq. (8) rewritten in terms of the velocity scale $\mathcal{U}^2 = 2/3\,k$ is

$$\frac{\mathrm{d}\mathcal{U}^2}{\mathrm{d}t} = -\frac{2}{3}\epsilon(t). \tag{10}$$

During the equilibrium decay the dissipation rate is described by the Taylor's law (1) and the Kolmogorov's type of the turbulence kinetic energy spectrum (3). For further comparisons with experimental data it is convenient to express the rate of change of velocity statistics as a function of the turbulence Reynolds number $Re = \mathcal{U}\mathcal{L}/\nu$. It is to note that the product of velocity and length scales $\mathcal{U}\mathcal{L}$ is proportional to the eddy viscosity $\nu_T$. Hence, the Reynolds number $Re$ in fact expresses the ratio of the eddy viscosity and the molecular viscosity $Re \sim \nu_T/\nu$. After substituting (1) into the RHS of Eq. (10) and further rearrangements we obtain

$$\frac{\mathrm{d}\mathcal{U}^{-2}}{\mathrm{d}t} = \frac{2}{3}C_\epsilon\frac{1}{\nu}\frac{\nu}{\mathcal{U}\mathcal{L}} = \frac{2}{3}C_\epsilon\frac{1}{\nu}\frac{1}{Re}. \tag{11}$$

To derive corresponding equation for the rate of change of $L$, the equilibrium law Eq. (1) is differentiated over time

$$\frac{1}{\epsilon}\frac{\mathrm{d}\epsilon}{\mathrm{d}t} = 3\frac{1}{\mathcal{U}}\frac{\mathrm{d}\mathcal{U}}{\mathrm{d}t} - \frac{1}{\mathcal{L}}\frac{\mathrm{d}\mathcal{L}}{\mathrm{d}t}. \tag{12}$$

To express the derivative $\mathrm{d}\epsilon/\mathrm{d}t$ the predictions of the classical $k-\epsilon$ turbulence model can be used (Launder and Sharma, 1974)

$$\frac{1}{\epsilon}\frac{\mathrm{d}\epsilon}{\mathrm{d}t} = C_0\frac{1}{\mathcal{U}^2}\frac{\mathrm{d}\mathcal{U}^2}{\mathrm{d}t}, \tag{13}$$

where $C_0 = 1.9$ is a model constant. Substituting (13) into (12) we obtain

$$\frac{1}{\mathcal{L}}\frac{\mathrm{d}\mathcal{L}}{\mathrm{d}t} = \left(\frac{3}{2} - C_0\right)\frac{1}{\mathcal{U}^2}\frac{\mathrm{d}\mathcal{U}^2}{\mathrm{d}t}. \tag{14}$$

After multiplying both sides by $2\mathcal{L}^2$, using Eqs. (1) and (10) the following relation is derived

$$\frac{1}{\nu}\frac{\mathrm{d}\mathcal{L}^2}{\mathrm{d}t} = A_e\frac{\mathcal{U}\mathcal{L}}{\nu} = A_e Re, \tag{15}$$

where $A_e = 4/3(C_0 - 3/2)C_\epsilon$. Hence, the classical theory predicts that the integral length scale increases with time during the decay of turbulence and that the decay of $\mathcal{L}^2$ slows down when the Reynolds number $Re$ decreases.

Even though $\mathcal{L}$ is expected to increase in time, the product $\mathcal{U}\mathcal{L}$ and also the turbulence Reynolds number $Re$ are expected to decrease in time. Indeed, after rearranging and combining Eqs. (11) and (15) we obtain the following formula:

$$\frac{\mathrm{d}\mathcal{U}\mathcal{L}}{\mathrm{d}t} = \mathcal{U}\frac{\mathrm{d}\mathcal{L}}{\mathrm{d}t} + \mathcal{L}\frac{\mathrm{d}\mathcal{U}}{\mathrm{d}t} = -\frac{1}{3}C_\epsilon\mathcal{U}^2 + \frac{A_e}{2}\mathcal{U}^2 = -\frac{1}{15}C_\epsilon\mathcal{U}^2 < 0. \tag{16}$$

## 2.3 Nonequilibrium decay

Scenario of a non-equilibrium decay predicts that the dissipation rate scales according to relation (2). After introducing this relation into Eq. (10) and further rearrangements we obtain

$$\frac{\mathrm{d}\mathcal{U}^{-2}}{\mathrm{d}t} = \frac{2}{3}C_{ne}\mathcal{U}_0\mathcal{L}_0\frac{1}{\nu^2}\left(\frac{\nu}{\mathcal{U}\mathcal{L}}\right)^2 = \frac{2}{3}C_{ne} Re_0\frac{1}{\nu}\frac{1}{Re^2}, \tag{17}$$

where $Re_0 = \mathcal{U}_0\mathcal{L}_0/\nu$. Equation (17) is different than the corresponding Eq. (11) derived for the equilibrium decay. It scales with $Re^{-2}$ instead of $Re^{-1}$ and, additionally, it depends on the initial conditions through $Re_0$. During the turbulence decay the Reynolds number $Re$ decreases with time, hence the ratio $Re_0/Re \geq 1$. This implies that decay rates predicted by Eq. (17) increase in time more sharply than those predicted by its equilibrium counterpart (11). Fast, anomalous changes of turbulence kinetic energy during non-equilibrium decay, which gradually decrease at later times, were observed experimentally by Meldi and Sagaut (2018).

As argued by Goto and Vassilicos (2016), in the initial stages of decay, after the forcing is stopped and long-range correlations suddenly disappear, the integral length scale starts to decrease with time. Using the self-similar form of the spectra (5) and relation (2), Steiros (2022b) derived the following formula for the time derivative of the length scale

$$\frac{1}{\nu}\frac{\mathrm{d}\mathcal{L}^2}{\mathrm{d}t} = A_{ne} - B_{ne}Re, \tag{18}$$

where $A_{ne}$ and $B_{ne}$ are coefficients related to the initial conditions and integrated spectral function. We note that the formula presented in the original paper by Steiros (2022b) was written in terms of the Reynolds number based on the Taylor length scale. However, in the non-equilibrium decay this length scale becomes proportional to the integral length scale $\mathcal{L}$, what follows directly from Eq. (2), see also discussion by Vassilicos (2015). Equation (18) is qualitatively different than its equilibrium counterpart (15). For large turbulence Reynolds numbers $Re$, the time derivative of $\mathcal{L}^2$ can be negative, causing $\mathcal{L}$ to decay in time. As the Reynolds number decreases during the decay of turbulence, the RHS of (18) will eventually become positive and $\mathcal{L}$ will start to increase with time until $\mathrm{d}\mathcal{L}^2/\mathrm{d}t$ reaches its maximum. At this point the system arrives at its equilibrium state and the statistics further follow equilibrium Taylor relation (1), however with a larger value of $C_\epsilon \approx 1$. This also implies that $\mathcal{L}$ further grows in time according to Eq. (15).

Results of numerical experiment presented in Steiros (2022b) confirmed that during the non-equilibrium decay the time derivative of $\mathcal{L}^2$ was a decreasing function of the turbulence Reynolds number. Moreover, in the initial stages of decay $\mathrm{d}\mathcal{L}^2/\mathrm{d}t < 0$.

## 2.4 Detection of nonequilibrium decay in ABL

Our purpose is to show, based on the experimental evidence, that the non-equilibrium form of decay is present in the atmo-
210 spheric turbulence before the sunset. During non-equilibrium decay, estimating dissipation rate from the low wavenumber part of the wind velocity spectra (3), or with the use of Eq. (1) becomes questionable and leads to underpredictions of the dissipation rate. On the other hand, resolution of the Doppler lidar is not sufficient to measure small turbulent motions, which are less affected by the non-equilibrium correction. Hence, unlike in laboratory experiments, direct verification of formulas (1) and (2) is not possible. For this reason, the non-equilibrium will be detected indirectly, by recording changes in the scaling of the
215 frequency spectra and the structure functions. According to relations (7) and (6), increasing deviations from the Kolmogorov scaling can be explained by the decrease of the integral length scale. This is in contrast to the theory of equilibrium decay where the integral length scale should increase with time according to Eq. (15) and the scaling of spectra and structure functions should become closer to the Kolmogorov one. We calculated both, frequency spectra and structure functions from time series of vertical velocity component measured by the Doppler lidar and estimated the scaling exponents (slopes) using least
220 squares fitting.

Apart from non-equilibrium correction, the slopes can be affected by insufficient resolution in time and space and high noise-to-signal ratio (Frehlich, 1994; Frehlich and Cornman, 1999). Banakh et al. (2021) investigated modifications of spectra due to instrumental noise, aliasing and due to space averaging. In particular, in the Fourier space the latter modification affects the whole range of scales and not only the highest wavenumbers. Moreover, the modification depends on the horizontal wind speed
$\bar{U}$. In order to convert time coordinate to space coordinate, Taylor's frozen eddy hypothesis is used, with $x = \bar{U}t$. This relation is justified only if the turbulence intensity defined as $\mathcal{U}/\bar{U}$, where $\bar{U}$ is the mean horizontal wind speed, is small enough. As $\bar{U}$ decreases, frequency spectra become more affected.

Because of these possible modifications of the spectra, in this work we focus on detecting changes of the slopes rather than on their exact values. We filter out data with high instrumental noise and data where $\mathcal{U}/\bar{U} < 0.15$. Although the frequency
spectra and structure functions are mathematically equivalent, they may respond differently to different sources of errors. In parallel to the slopes, we present the calculated integral length scales and mean wind velocity to verify whether the changes of the scaling are due to changes of the integral length scale, as predicted by relations (7) and (6) or, rather are affected by changes of the mean wind speed.

## 3 Experimental sites, meteorological conditions and instrumentation

The data used for this work were obtained using the Doppler Lidar system in rural and urban environments. The measurements were performed over a rural environment during POLIMOS (Technical assistance for Polish Radar and Lidar Mobile Obser-

vation System) campaign, which took place between May and September 2018 at PolWET peatland site in Rzecin (52°45'N, 16°18'E, 54 m a.s.l.), Poland. The measurements over an urban environment performed at Warsaw Observation Station in the center of Warsaw (52°12'N, 20.°58'E, 112 m a.s.l.), Poland. Both sites are part of the Aerosol, Cloud and Trace Gases Infrastructure (ACTRIS-ERIC). The locations for each sites are presented on Fig 1.

During the measurements the meteorological conditions were represented by the hot and dry periods for each of the sites. In 2018, the meteorological conditions in Rzecin deviated from the reference values, for precipitation and mean air temperature (512 mm and 8.63°C, respectively), with recorded values of 464 mm and 9.63°C (Poczta et al., 2023). Furthermore, the summer of 2018 was one of the hottest and driest periods over recent years (122 mm and 19.21°C) in comparison to the reference values (192 mm and 18.0°C) for this season. The meteorological conditions in 2023 in Warsaw also differ from the reference values for precipitation and mean air temperature (549.7 mm and 9.00°C respectively), with recorded values of 620.9 mm and 11.07°C. Even though the total amount of precipitation for 2023 in Warsaw was higher than the reference value, the summer of 2023 was drier and hotter (184.4 mm and 20.29°C) than the reference values (257.1 mm 17.65°C) for this season. A relative increase in temperature (+14.96% and +6.72% in comparison to the reference values respectively for Warsaw and Rzecin) and relative decrease of precipitation amount (-28.28% and -36.46% in comparison to the reference values respectively for the Warsaw and Rzecin) was observed for both of the measurement locations during summer seasons. It follows that the meteorological conditions for both stations were relatively similar and were characterized by higher temperatures and less amount of precipitation for the summer season both for 2018 and 2023 in comparison to the historical data. The reference values, for precipitation and mean air temperature for the Rzecin were calculated based on the Szamotuły-Baborówko station meteorological data (IMGW-PIB, 2024a), for period between 1990-2014 (further data not available). The reference values for Warsaw were calculated from the "Climate Standards 1990-2020" (IMGW-PIB, 2024b). The amount of precipitation and average air temperature value for the summer season 2023 in Warsaw were calculated based on the Warszawa-Filtry station meteorological data (IMGW-PIB, 2024a).

For each location, the vertical and horizontal wind profiles were obtained using the measurements from Streamline (Halo Photonics) Doppler lidars. In rural environment, the measurements were taken using the Doppler lidar provided by the Research Atmospheric Physics Group from the University of Granada (GFAT-UGR). The Doppler Lidar operating in Warsaw is owned by the Remote Sensing Laboratory (RS-Lab), at the Faculty of Physics of the University of Warsaw. The lidars comprise of a solid-state pulsed laser emitting at 1.5 $\mu$m and a heterodyne detector with fiber-optic technology. The emission is provided with the pulses of energy at 100 $\mu$J, pulse duration of 200 ns and pulse repetition rate of 15 kHz and 10 kHz respectively for GFAT-UGR and RS-Lab systems. The signal acquisition is done in continuous and autonomous vertical mode and regular measurements are done in the vertical display azimuth (VAD). A more detailed description of the Doppler Lidar system can be found in Pearson et al. (2009).

The lidar signal acquisition was performed continuously in vertical mode obtaining a vertical wind component with a 30 m of spatial and 1 s of temporal resolution. For the horizontal profiles of the wind, the Vertical Azimuth Display (VAD) scans with a constant elevation of 70° and 12 azimuth points, were performed every 30 minutes. The focus of the optical system was at the 535 ± 35 m (Ortiz-Amezcua et al., 2022) and at the infinite value respectively for GFAT-UGR and RS-Lab lidars. To support

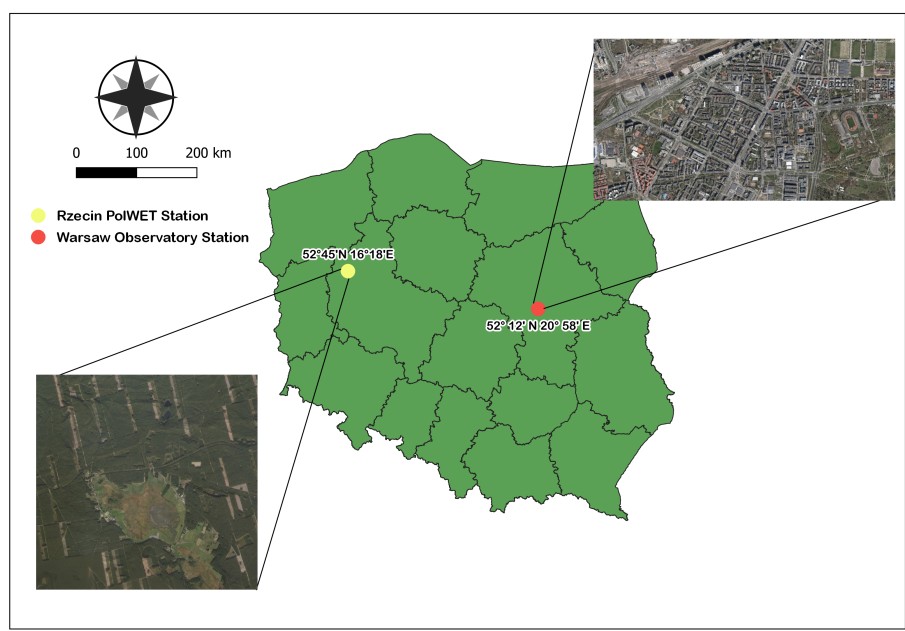

**Figure 1.** Location of ACTRIS-ERIC experimental sites, Rzecin PolWET Station of Poznan University of Life Sciences and Warsaw Observatory Station of Univeristy of Warsaw. The map was provided by Zuzanna Rykowska (University of Warsaw).

our findings we additionally analyzed the momentum and heat fluxes (30 min. averages). The eddy covariance observations were performed in parallel to the lidar measurements with instruments mounted on a meteorological tower (4.5 ma.g.) in the Rzecin PolWET Station (see Poczta et al. (2023) for details) and in the Radiation Transfer Laboratory measuring platform on the roof of the building of the Faculty of Physics in Warsaw (22ma.g.).

## 4 Methodology

The whole database of Doppler Lidar's and surface flux measurements consists of 4 months (June - September 2018) of measurements in a rural environment and 4 months (June - September 2023) in an urban environment. To obtain the data ready for further analyses the raw Doppler lidar data was firstly background corrected, using the calibration procedures proposed by Manninen et al. (2016) and Vakkari et al. (2019), and secondly filtered out by values of signal to noise ratio treshold (SNR = 1.006) (Manninen et al., 2016). Errors of the vertical and horizontal velocity measurements were calculated using the software processing HALOlidar toolbox (Manninen, 2019), with methods proposed by Rye and Hardesty (1993) and Pearson et al. (2009) for the vertical components and by Päschke et al. (2015) for the horizontal ones.

The lidar data were first used to estimate the ABL height. We compare results of two methods. The first one is the gradient method of the backscatter signal. In lidar measurements, the backscatter signal in the ABL is significantly stronger than in

the free troposphere so there is a distinct change in backscatter signal values when it passes through the boundary between the ABL and the the free troposphere (Wang et al., 2020; Liu et al., 2022). The boundary layer height is defined as the height where the range corrected signal sharply decreases, that is, where the its gradient has a minimum value (Dang et al., 2019). The second, variance method was used to estimate the height of the Convective Boundary Layer (Dewani et al., 2023). The top of the layer is estimated as the height, where the variance of the vertical velocity decreases below a certain threshold value. We use the threshold of 0.09, as in Dewani et al. (2023).

To compute the slopes of the frequency spectra and structure functions, and values of integral length scales the vertical velocity measurements at different heights were grouped in 0.5 h intervals. Recorded signal was decomposed into the mean and fluctuating part as

$$w' = w - \langle w \rangle, \tag{19}$$

where $\langle w \rangle$ is a 600s running average. This detrending removes the largest, convective scales from outside the inertial sub-range and allows for a better convergence of statistics within 0.5 h intervals. Turbulence velocity scale was calculated from the vertical velocity fluctuations as

$$\mathcal{U} = \langle w'^2 \rangle^{1/2} \tag{20}$$

for each height and time interval. Cases where the turbulence intensity was larger than 0.15, which would not support Taylor's hypothesis, were filtered out. The frequency spectra were computed for each group and a logarithmic fit in the frequency range $f \in [0.15\mathrm{s}^{-1}, 0.3\mathrm{s}^{-1}]$ was performed without fixing the slope. In the logarithmic plot, the power-law function forms a straight line, and its slope is equal to the scaling exponent. We calculated the slopes at each height and each time interval using the least squares algorithm and investigated whether the slopes deviate from the Kolmogorov's predictions (-5/3 and 2/3 scaling of the frequency spectrum and structure function, respectively). To determine the integral length scale we first calculated the two-point transverse correlation coefficient

$$g(r) = \frac{\langle w(x)w(x+r) \rangle}{\langle w(x)^2 \rangle}. \tag{21}$$

According to the theory of homogeneous, isotropic turbulence (Pope, 2000), the function $g(r)$ takes the form

$$g(r) = \exp\left(-\frac{r}{\mathcal{L}}\right)\left(1 - \frac{1}{2}\frac{r}{\mathcal{L}}\right). \tag{22}$$

and crosses the horizontal axis at $r = 2\mathcal{L}$. We numerically integrated the function $g(r)$ from $r = 0$ to $r = 2\mathcal{L}$. This integral should be approximately equal to $0.57\mathcal{L}$.

Chart 2 presents the methodology to obtain mean horizontal wind, slopes of frequency spectrum and structure function, and integral length scale for each interval. For slopes we additionally use the $R^2$ threshold (O'Connor et al., 2010), such that fits with $R^2$ less than 0.6 are considered noise and discarded.

To calculate the buoyancy flux $\langle w'b' \rangle$ defined in Eq. (9) using the data from PolWET station we approximated the virtual potential temperature as

$$\theta_v = \theta \left[ 1 + \left( \frac{R_v}{R_d} - 1 \right) q_v \right] \approx \theta \left[ 1 + 0.61 q_v \right], \tag{23}$$

where $R_v$ and $R_d$ are gas constants for water vapor and dry air, $q_v$ is the mixing ratio of water vapour and we assumed $\theta \approx T$, where $T$ is absolute temperature. With this the buoyancy flux was calculated as

$$\langle w'b' \rangle = \frac{g}{\langle \theta_v \rangle} \langle w'T' \rangle + 0.61\, g \langle w'q_v' \rangle. \tag{24}$$

For the Warsaw data we approximated the virtual potential temperature with the measured sonic temperature $\theta_v \approx T_s$ and calculated the buoyancy flux as

$$\langle w'b' \rangle = \frac{g}{\langle T_s \rangle} \langle w'T_s' \rangle. \tag{25}$$

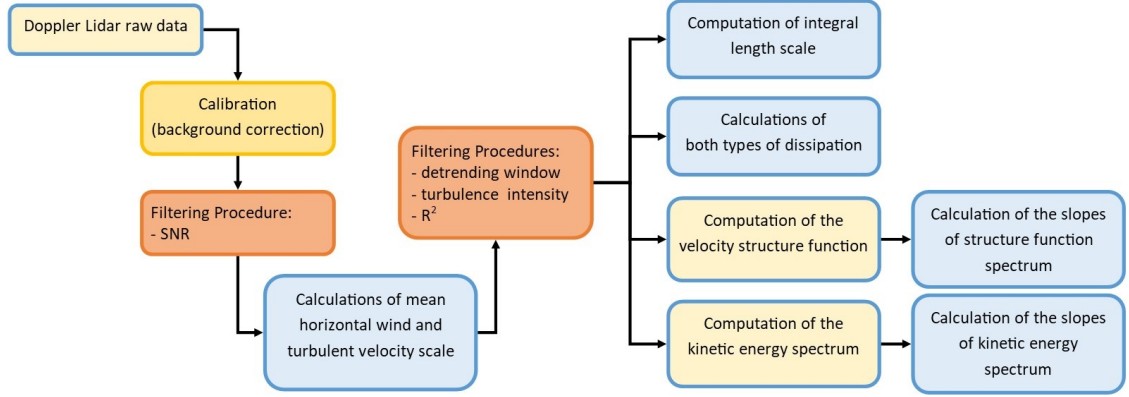

**Figure 2.** Methodology chain of obtaining turbulence properties: calibration (orange), filtering procedures (red), calculating turbulence properties (blue), remaining procedures (yellow).

## 5    Results

### 5.1    ABL height, velocity and time scales

As in this work we focus on the evening hours, we denote the sunset time as $t = 0$. Turbulence statistics are calculated as 1/2 hour averages or medians relative to the sunset time both for Rzecin and Warsaw for the same months during the summer. We present the statistics as functions of time to analyze how they change during rapid decay of turbulence.

We first present the evolution of the heights of ABL and Convective Boundary Layer (CBL) in Fig. 3. The uncertainties
were estimated as a standard error of the mean (SEM) of the half-hour time intervals, see Appendix A. As it is observed, the
ABL height at the rural site is lower and decreases more rapidly compared to the urban site. This may indicate the influence
of the urban heat island effect in large cities. The behavior of the CBL height is similar, i.e. much higher values are observed
in Warsaw. Both in Rzecin and Warsaw sites the CBL collapses rapidly ca. 2.5 hours before the sunset. The results are in
agreement with the long-term study by Wang et al. (2020).

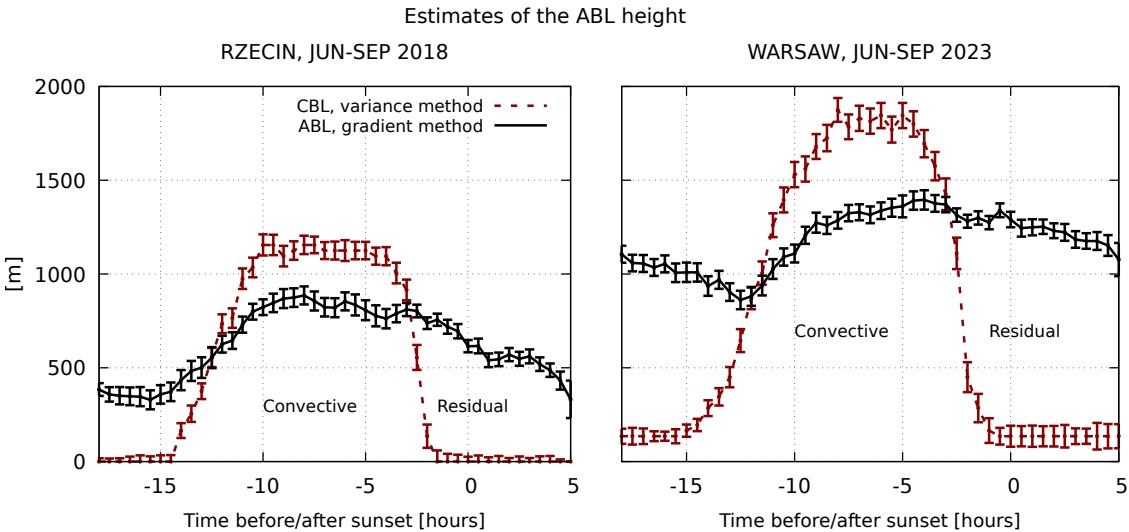

**Figure 3.** The median values of the ABL and CBL heights and error estimates (see Appendix A) during summer season (June-September) at
Rzecin PolWET station on 2018 (left column) and Warsaw Observatory Station on 2023 (right column).

In order to estimate the beginning of the evening transition we present the buoyancy fluxes, their median values and standard
errors in Fig. 4. At $t = -2$h the median values of the fluxes are still positive. Afterwards they cross the zero-level in Rzecin
site and become close to zero in Warsaw site. The buoyancy fluxes and the CBL height are further used to calculate the char-
acteristic scales which govern the flow during daytime convection. The convective Deardorff scale $w_*$ and the corresponding
time scale $\tau$ are defined as (Deardorff, 1970)

$$w_* = (D\langle w'b'\rangle_0)^{1/3}, \quad \tau = \frac{D}{w_*}, \tag{26}$$

where the CBL height was denoted by $D$ and $\langle w'b'\rangle_0$ is the surface value of the buoyancy flux. We present both scales in Fig.
5 and additionally compare $w_*$ with the friction velocity $u_*$. Description of the error estimates is given in Appendix A. As it is
seen in Fig. 5, both in Rzecin and in Warsaw $w_*$ is still larger than $u_*$ at $t = -2$h and the time scale $\tau$ increases sharply around
$t = -2$h. This time scale is larger in Warsaw, which suggests that the turbulence decay is slower in the urban surface layer.
Shortly after $t = -2$h, $u_*$ becomes larger or comparable to $w_*$ (within the confidence intervals), which implies that turbulence
production due to shear $P$ becomes dominant in the surface layer. We will assume that at larger altitudes the contribution of

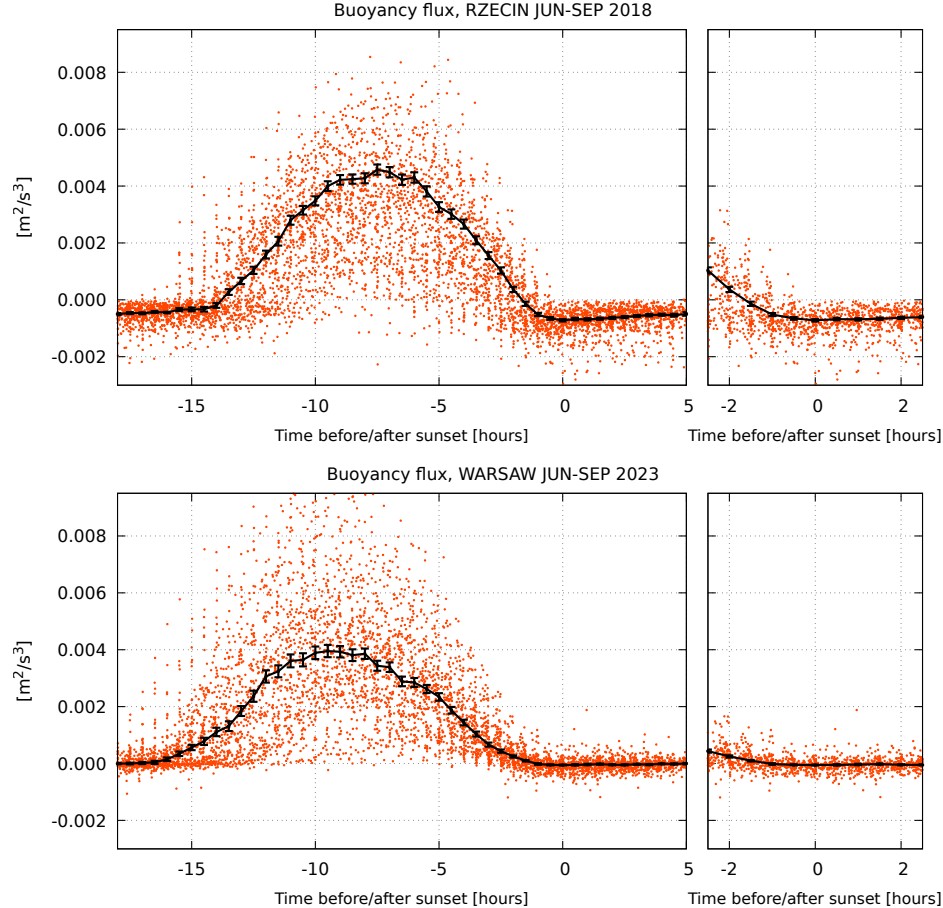

**Figure 4.** The median values of the bouyancy flux and error estimates (see Appendix A) during summer season (June-September) at Rzecin PolWET station in 2018 (upper) and Warsaw Observatory Station in 2023 (lower).

the shear production is negligible and that to the leading order the evolution of the turbulence kinetic energy is described by Eq. (10). Under such assumptions the changes of $\mathcal{U}$ and $\mathcal{L}$ should be described by formulas derived in Sections 2.2 and 2.3. For this reason we will further treat statistics at $t = -2\mathrm{h}$ as initial conditions and focus our attention on the time interval $\pm 2\mathrm{h}$

relative to the sunset.

## 5.2 Spectra and structure functions

At 18:30 UTC, with convection still present, the wind energy spectrum was measured at a height of 195 $\mathrm{ma.g.l.}$, i.e. within the mixed layer, (cf. Fig. 6a). In this case the spectrum is visibly steeper than -5/3. The steep slopes of frequency spectra in the convective regime were also reported by other authors (Darbieu et al., 2015). Xie and Huang (2022) speculated they can

be linked to the presence of inverse cascades at large scales, which leads to the $-11/5$ Bolgiano-Obukhov scaling (Bolgiano,

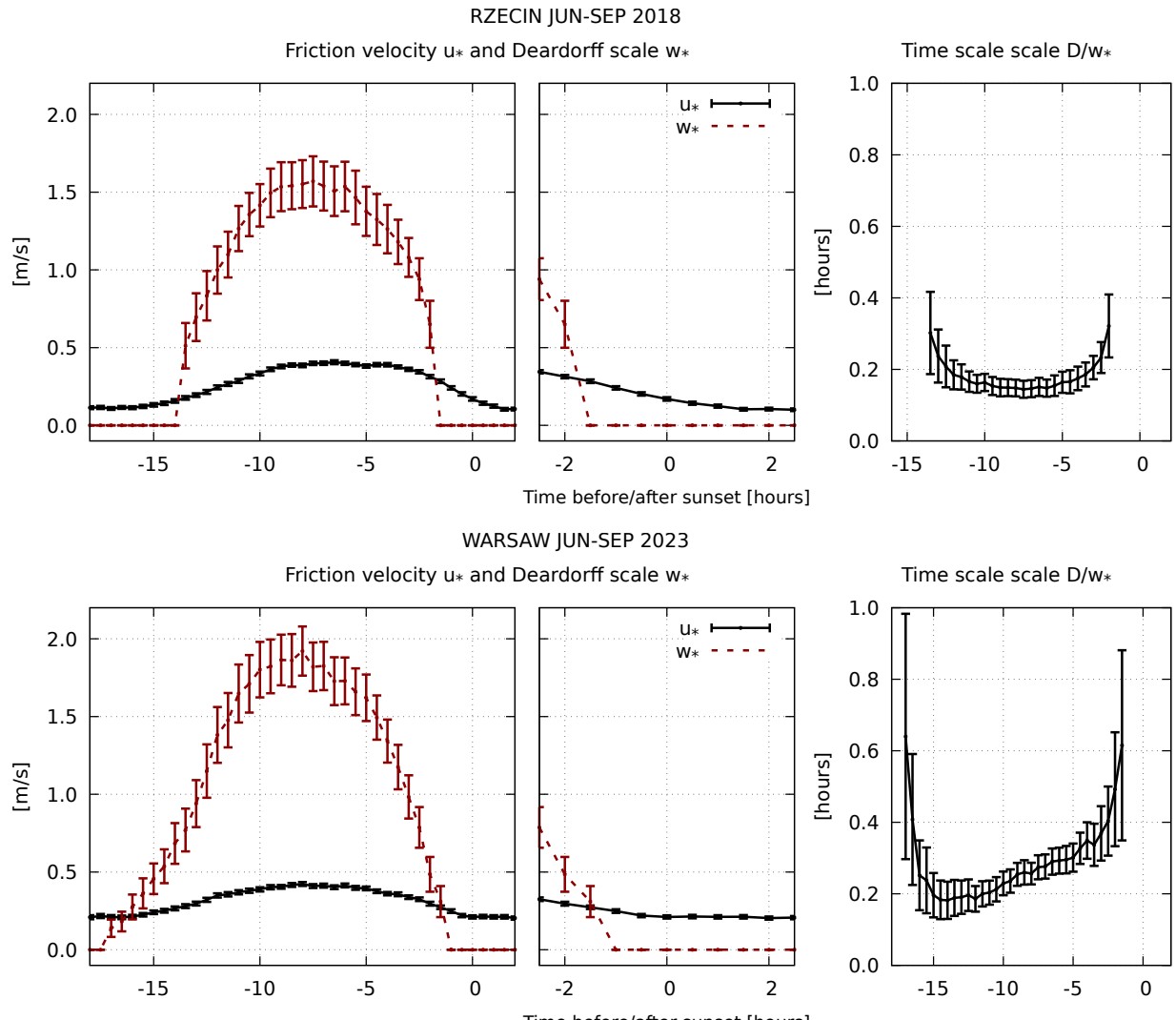

**Figure 5.** The mean friction velocity $u_*$, convective Deardorff scale $w_*$, the time scale $\tau$ and error estimates (see Appendix A) during summer season (June-September) at Rzecin PolWET station in 2018 (upper figures) and Warsaw Observatory Station in 2023 (lower figures).

1959). However, as argued by Banakh et al. (2021), in case of poorly resolved data the steepening of slopes may also be caused by the artificial, instrumental dissipation due to effective low-pass filtering.

Figure 6b shows that before the sunset, when the convective layer rapidly decays, the frequency spectrum becomes less steep than the Kolmogorov's prediction. This observation is not related to instrumental artificial dissipation, which rather cause
opposite effect (steepening of the spectra).

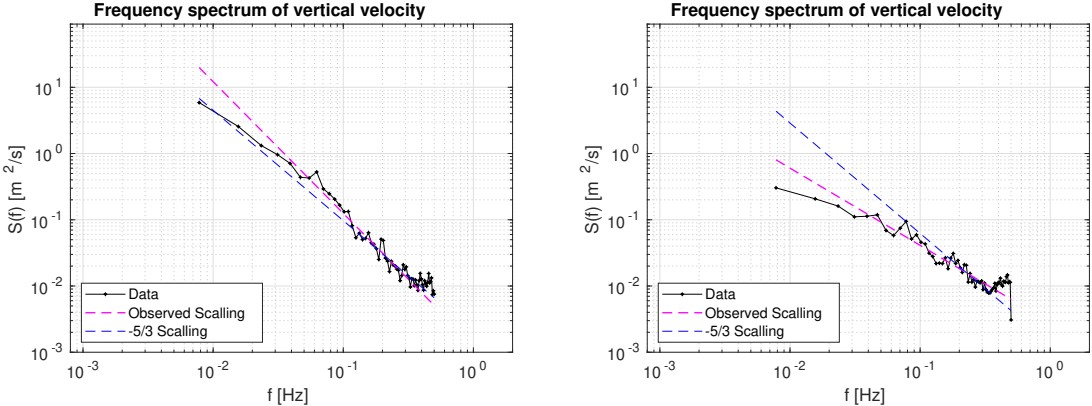

**Figure 6.** Exemplary frequency spectra of vertical wind measured on 28.06.2018 at Rzecin PolWET station, at a) 195 m.a.g.l. at 18:30 (UTC) and at b) 195 m.a.g.l. at 19:30 (UTC) showing, respectively, the steeper and less steep frequency spectrum as compared to the Kolmogorov's prediction.

In Figure 7, time-height evolution of the slopes of the frequency spectra and velocity structure functions and the estimate of the CBL height are shown. The regions marked by red and orange colours in Fig. 7a and yellow and light-green in Fig. 7b exhibit the scaling steeper than Kolmogorov. Before the sunset, when the CBL collapses, the turbulence kinetic energy decays rapidly (El Guernaoui et al., 2019) and a sharp decrease of the slopes at all heights is observed. Decrease of the slopes is also
seen in the upper part of the convective ABL, where the raising updrafts become weaker. Therein, stable stratification possibly alters the spectra and structure functions. The stratification effects on spectral slopes were included in a recent model by Cheng et al. (2020). This issue is, however, beyond the scope of the present work, as we rather focus on the modification of the spectra due to non-stationarity.

## 5.3 Influence of detrending window

We next investigated how different values of the detrending window influence the results, in order to choose the most proper value. The size of detrending window can potentially affect the turbulence velocity scale and the integral length scale, calculated from Eqs. (20) and (22), respectively. To estimate the errors we took into account the standard error of the mean and errors of velocity estimates from the HALOlidar toolbox (see Appendix A for details). We calculated and compared both mean and median values and compared them in Fig. 8. Median values are advantageous if the probability distribution of data is non-
Gaussian or in the presence rare but very large or very small values (outliers), which affect the mean. As can be seen in Fig. 8 all the mean values are larger then the corresponding medians. Both, the mean and median values of $\mathcal{U}$ are not much affected by the change of the detrending window in the algorithm. They decrease with time during the evening transition. Mean and median values of the integral length scales increase with increasing size of the detrending window. The data become considerably scattered for the largest window. This is to be expected, as $\mathcal{L}$ converges slowly and a large sample is needed to
estimate it with sufficient accuracy (Lenschow et al., 1994). The time span needed to calculate statistics is proportional to $\mathcal{L}$,

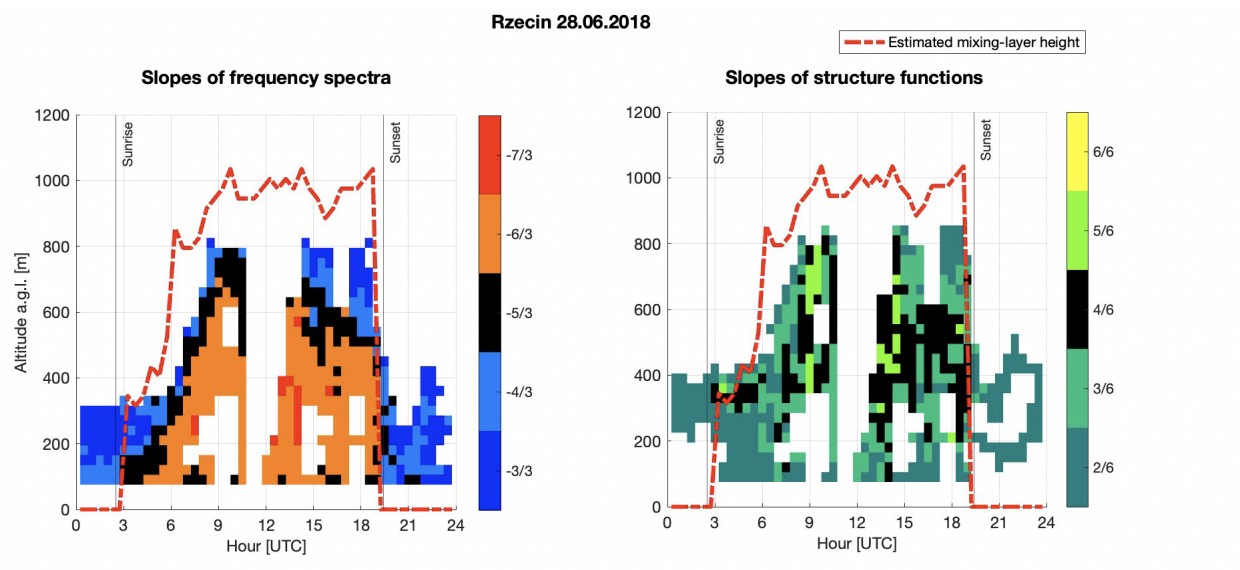

**Figure 7.** Slopes of the frequency spectra and velocity structure functions and estimates of the CBL height, for data measured on 28.06.2018 at Rzecin PolWET station. The boxes marked by colour black indicate the slopes of frequency spectra equal to -5/3±1/6 and slopes of velocity structure function equal to 2/3±1/12.

hence larger length scales require longer time spans. As a result, 1/2 hour averages may be insufficient to reduce random errors to acceptable levels needed to calculate the time derivatives from Eqs. (15) and (18). Independent of the size of the detrending window, we find that the median values of integral length scales first decrease with time and next increase or become constant. This conclusion seems to be universal and allows further analysis to be performed only using one set value for the detrending window, which has been chosen to be 600 s. The median values of the product $\mathcal{UL}$ (and also the turbulence Reynolds number $Re$) decrease with time, as expected, for all detrending windows. The corresponding mean values decrease in time before the sunset for the two smaller detrending windows.

### 5.4 Turbulence properties: statistical analysis

To investigate how the turbulence properties change not only in time but also with altitude we divided results into 3 altitude ranges: 75–345m.a.g.l., 345–585 m.a.g.l., 585–855 m.a.g.l. According to Fig. 3, the measurements are within the residual layer and in Rzecin the level 75–345m.a.g.l. corresponds to the lower and middle part of the ABL, 345–585 m.a.g.l. reaches the top of the layer and the third level 585–855 m.a.g.l. is placed partly above the mean top of the ABL. The ABL is higher in Warsaw, such that the three levels corresponds to its lower, middle and upper part, respectively. In spite of the scatter of results for individual 1/2 hour averages, median values of the slopes of frequency spectra determined for the range $f \in [0.15\mathrm{s}^{-1}, 0.3\mathrm{s}^{-1}]$ (Fig. 9) clearly increase from values close to Kolmogorov $-5/3$ to around $-1$ after the sunset. Analogously, slopes of the structure functions (Fig. 10) determined for the range $r \in [40\mathrm{m}, 150\mathrm{m}]$ decrease with time for both sites in Rzecin

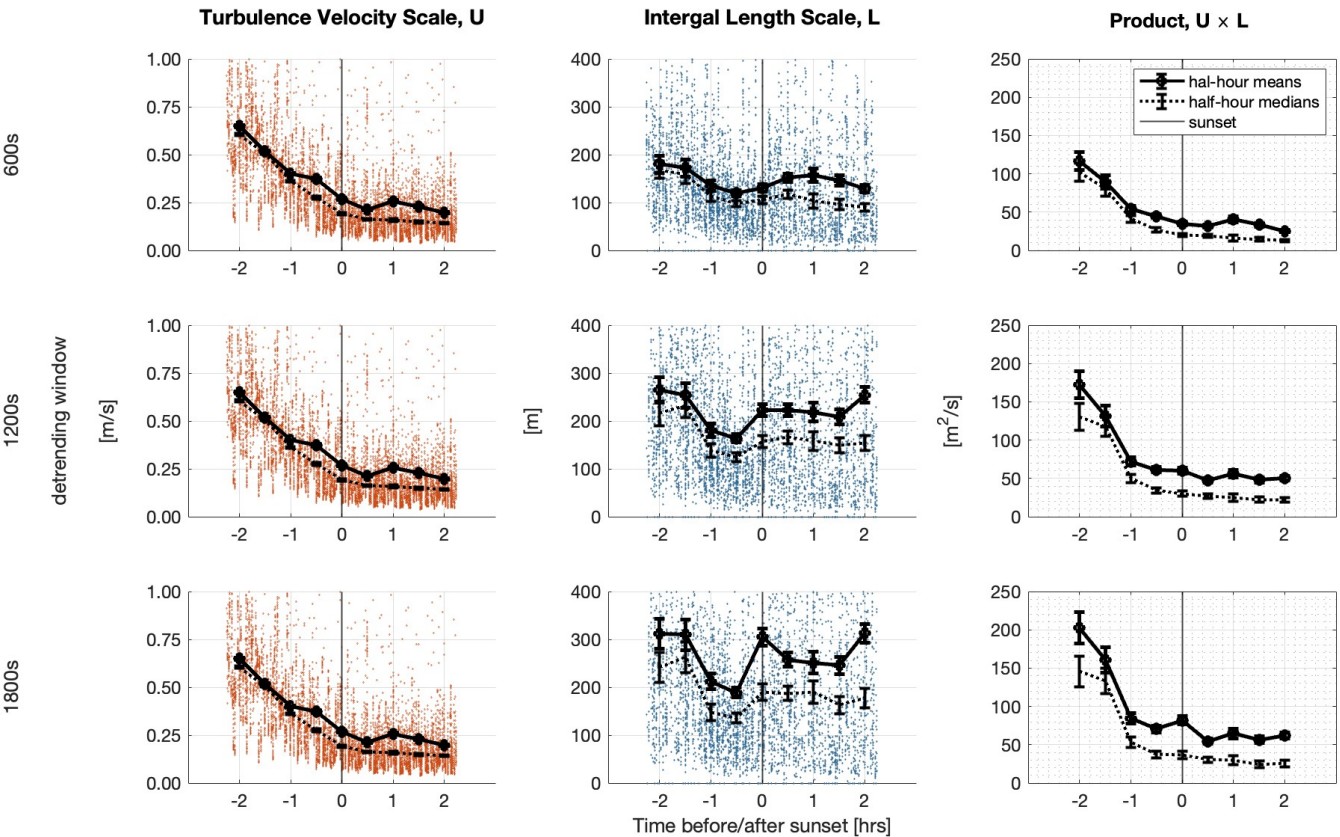

**Figure 8.** Mean and median values of the turbulence velocity scale $\mathcal{U}$, the integral length scale $\mathcal{L}$ and the product $\mathcal{U}\mathcal{L}$ and error estimates (see Appendix A) for different detrending windows, for data measured at Rzecin PolWET station on August 2018.

and Warsaw. The inertial ranges of structure functions tend to be smaller than those for wind frequency spectra. Hence, due to finite temporal resolution of the measurements, the calculated structure functions may be affected by large eddies, from beyond the inertial range. As a result, the slopes of structure functions are gentler than the $2/3$ Kolmogorov predictions even at $t = -2$h, especially for the Rzecin site. The slopes also show the altitude-dependent relation. What can be observed from both figures (Fig. 9 and Fig. 10), is that for Warsaw the median values of slopes at higher altitudes are much closer to the Kolmogorov's predictions than for the rural case. It can be partly explained by the urban heat island effect and differences between the urban and rural morphologies. The mechanical turbulences are generated due to an intense shear at the top of the canopy layer $H$ (Roth, 2000). In the urban environment it is at the height of the rooftops of the buildings. In Warsaw, on a 1 km × 1 km area centred in the RS Lab $H \approx 12.2$m. In Rzecin, the top canopy layer height reaches $H = 2$–$2.5$m. As far as the thermal-driven turbulences are concerned, the city centres are usually warmer than the suburbs and agglomeration, because of

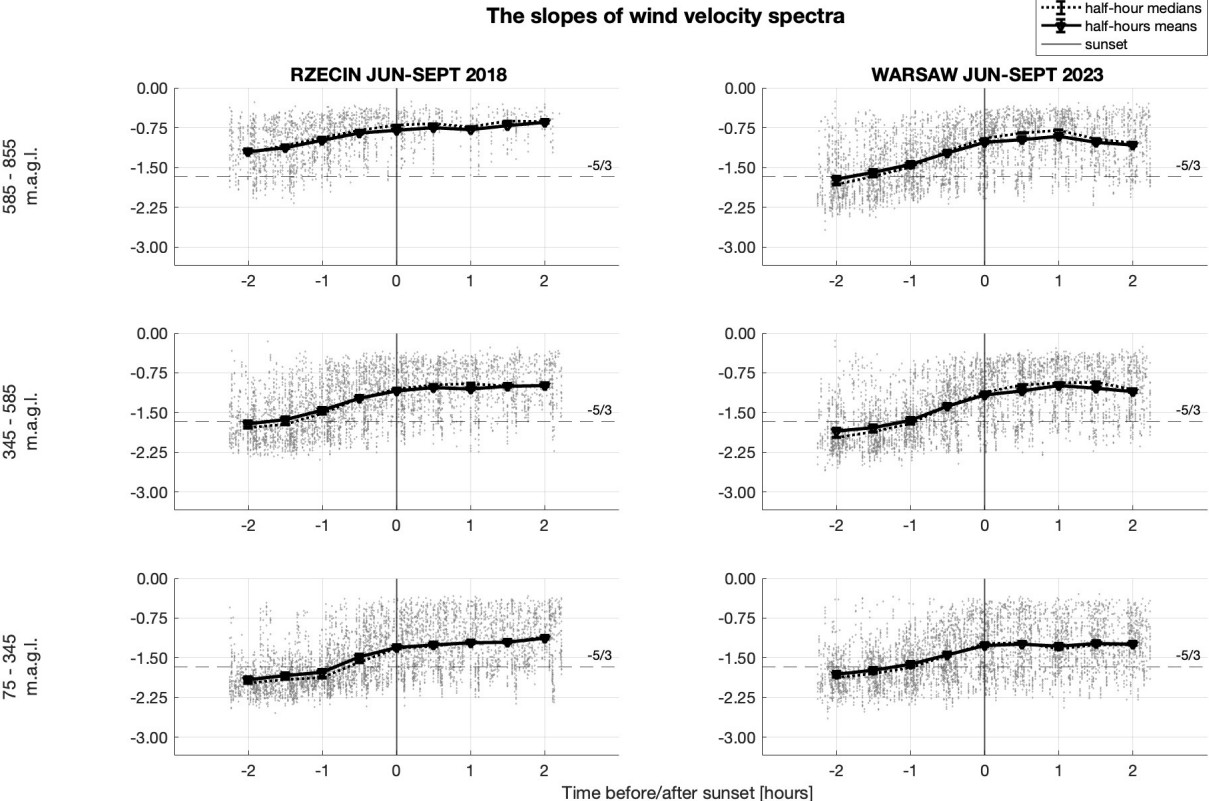

**Figure 9.** Slopes of the frequency spectra of vertical wind, mean, medians and and error estimates (see Appendix A) at different altitudes for data measured during summer season (June-September) in, respectively, rural and urban environment at Rzecin PolWET station on 2018 (left column) and Warsaw Observatory Station on 2023 (right column).

the urban heat island phenomenon (Kuchcik et al., 2014; Stopa-Boryczka et al., 2002), and related to this, relatively higher heat capacity (Oke, 1982) compared to the rural environment. This implies that the urban surface can still be able to emit heat, even after sunset. Heat emitted from a warm urban surface generates convection and mixes the air in the urban canopy layer. It also generates a dome of warm air in higher parts of the boundary layer. The temperature profile of this dome is quasi-adiabatic and similar to the temperature profiles during a midday (Oke, 1987). Results presented in Figures 9 and 10 suggest, however that the heat island effects are most significant 2 hours before the sunset at the highest altitude range 585–855 m.a.g.l. Afterwards, a rapid change of the scaling exponent is observed.

In order to examine if the changes in scaling exponent are induced by changes in the mean velocity, we analyzed how $\bar{U}$ changes with time and height. If turbulence were Kolmogorov-like, and the spectra were affected only by spurious modifications due to insufficient resolutions in time and space, then a decrease of mean velocity would increase the absolute value of the slopes by introducing an artificial dissipation. Increase of the mean velocity, on the other hand would bring the scaling closer

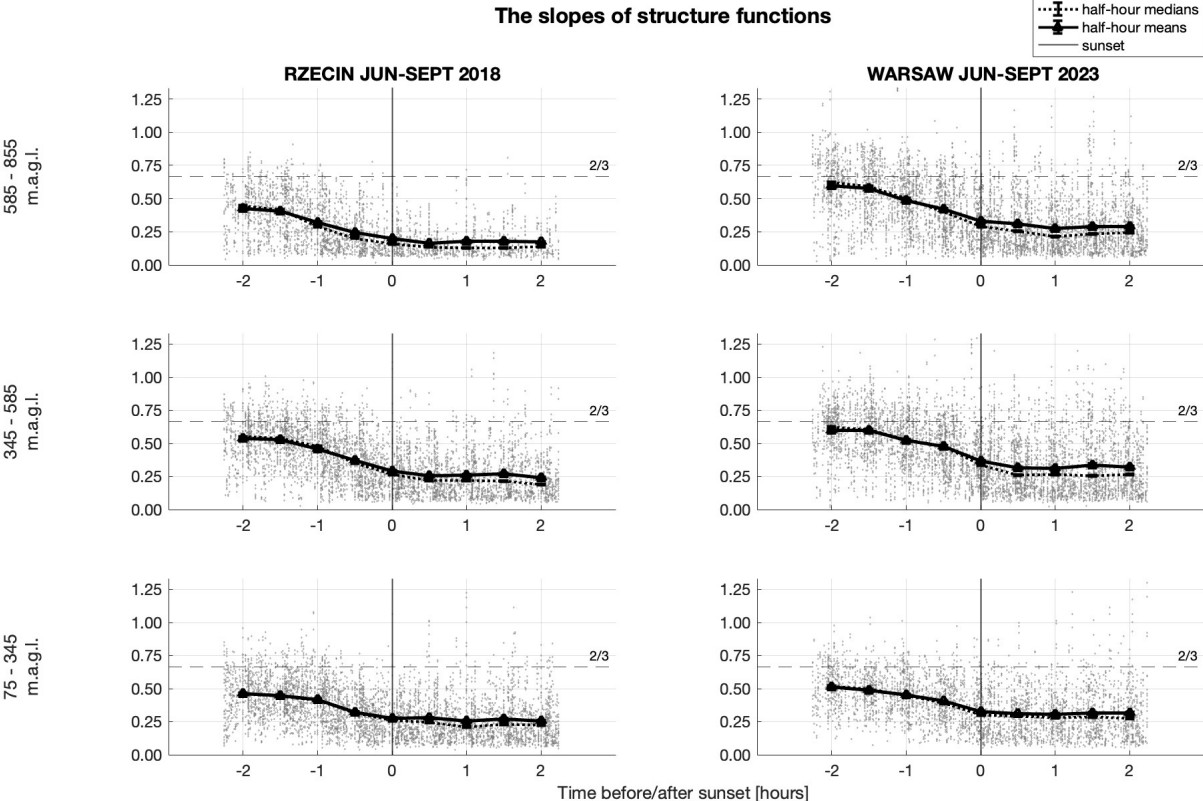

**Figure 10.** Slopes of the second-order structure functions,  mean, medians and error estimates (see Appendix A)  at different altitudes for data measured during summer season (June-September) in, respectively, rural and urban environment at Rzecin PolWET station on 2018 (left column) and Warsaw Observatory Station on 2023 (right column).

to the Kolmogorov 5/3 (or 2/3 for the structure function), but not below this value. Figure 11 presents the mean horizontal velocity $\bar{U}$. As is is seen, $\bar{U}$ increases with time mostly close to the surface and only after the sunset. At higher altitudes it

has only a slight tendency to increase. At the same time, the absolute values of slopes in Figs. 9 and 10 decrease considerably below 5/3 and 2/3, respectively. Hence, we conclude that changes in the slopes are not primarily affected by the changes in the mean wind speed. Instead, the collapse of the largest convective motions possibly leads to the non-equilibrium states of turbulence as predicted by Eqs. (6) and (7). Later on, the size of the inertial range decreases and is shifted towards small scales (large wavenumber) which are not detected by the lidars.

A difference between rural and urban sites observed in Fig. 11 is that in the latter, mean velocity starts to slightly increase before the sunset at low altitudes. As described by Mahrt (2017), when turbulence decreases rapidly, the airflow becomes more influenced by the surface heterogeneity and horizontal temperature variations. The temperature variations lead to the local

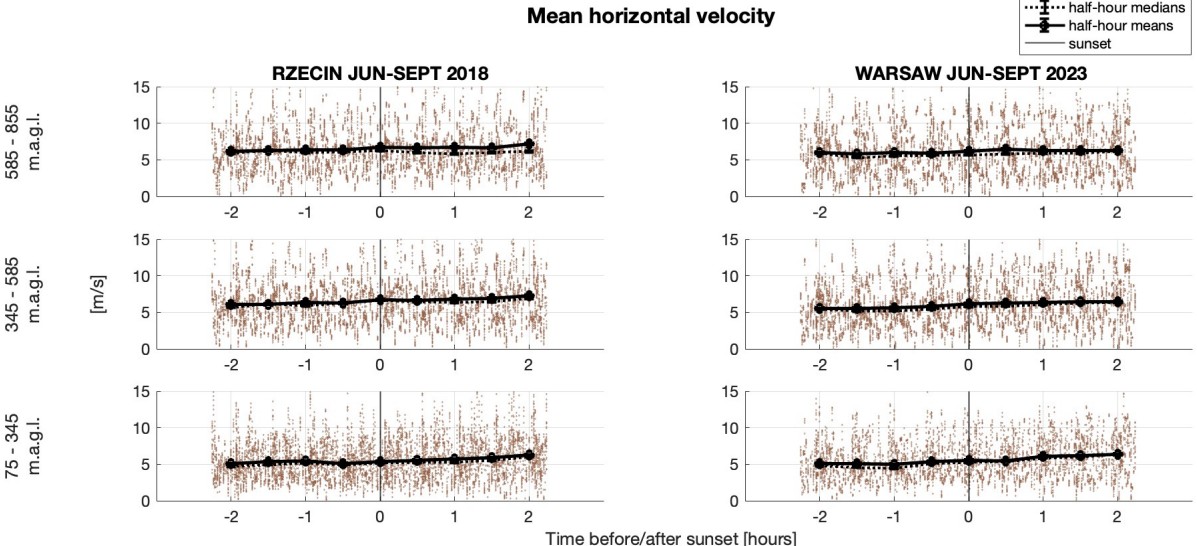

**Figure 11.** Horizontal velocity, mean, medians and error estimates (see Appendix A) at different altitudes for data measured during summer season (June-September) in, respectively, rural and urban environment at Rzecin PolWET station on 2018 (left column) and Warsaw Observatory Station on 2023 (right column).

horizontal pressure gradients. These, in turn could induce stronger horizontal winds to increase the horizontal temperature transport.

Mahrt (1981) discussed a formation of early evening calm periods, during which the mean velocity decreased in the surface layer while it had a tendency to increase at higher altitudes. Busse and Knupp (2012) on the other hand, observed the decrease of the mean speed at altitudes, up to 500m. The early evening calm periods in the surface layer were also recorded in other studies (Mahrt et al., 2012; Román-Cascón et al., 2015). No systematic decrease of the mean wind speed is observed in Fig. 11 before the sunset. However, we recall that to calculate turbulence properties we removed data for which the Taylor frozen

eddy hypothesis was not satisfied, that is data with high $\mathcal{U}/\bar{U}$ ratio. This procedure could to some extent, affect the tendencies observed in Fig. 11.

    The dependence of the integral length scales on time and altitude is presented in Fig. 12. The vertical integral length scales increase with the altitude. Moreover, at the urban site they are larger at higher altitudes, which follows from stronger convection and more shear-driven turbulences at this part of ABL. As argued by Akinlabi et al. (2022) the roughness surface layer in cities

may be higher than previously expected and can reach up to $z/H = 30$, where $H$ is the mean height of buildings (in Warsaw site $H \approx 12.2$m).

    According to Fig. 13, the turbulent velocity scale is decreasing with time for both rural and urban cases. In connection with the stable horizontal wind velocity, before and after the sunset it shows that the observed turbulences are mostly driven by

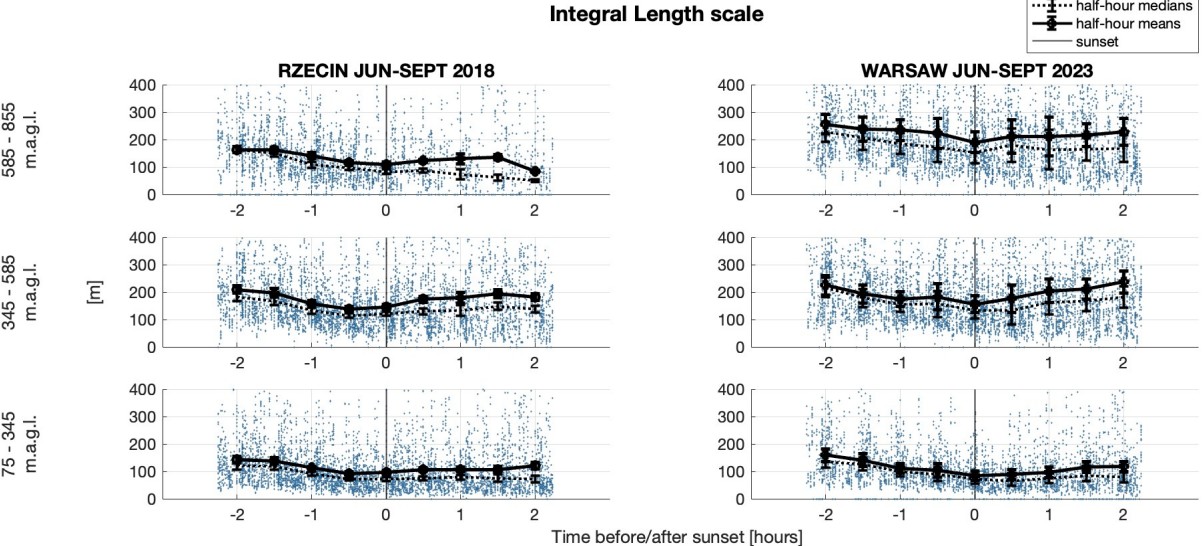

**Figure 12.** Integral length scales at different altitudes for data, mean, medians and error estimates (see Appendix A), measured during summer season (June-September) in, respectively, rural and urban environment at Rzecin PolWET station on 2018 (left column) and Warsaw Observatory Station on 2023 (right column).

the convection, which agrees with our observational statement about the decay of the ABL and the presence of the decaying

convection-driven turbulences before the sunset.

To study decay of turbulence with reference to the values measured 2h before the sunset, normalised 15 min median values $\mathcal{L}/\mathcal{L}_0$ and $\mathcal{U}/\mathcal{U}_0$, where $\mathcal{L}_0 = \mathcal{L}(t=-2\text{h})$, $\mathcal{U}_0 = \mathcal{U}(t=-2\text{h})$ are plotted in Fig. 14. In this figure we included only data measured before the sunset. It is seen that both turbulence kinetic energy and the integral length scale have a tendency to decrease with time before the sunset. As follows from Eqs. (6) and (7), decrease of the integral length scale during the decay

of turbulence will cause increasing deviations from the Kolmogorov scaling. This appears to be the case in the observed decay of convective boundary layer and explains the changes of the slopes, presented in Figs. 9 and 8.

We finally calculated the time derivatives $d\mathcal{U}^{-2}/dt$ and $d\mathcal{L}^2/dt$ from the 1/2 h median values (for both sites and at all heights) for $-2\text{h} \leq t \leq 0$ (evening hours) and presented them in Fig. (15) as functions of $Re$. We compare experimental data with the classical predictions (11) and (15) with $C_\epsilon = 0.5$ and the non-equilibrium predictions (17) and (18). The coefficients

$A_{ne}$, $B_{ne}$ and $C_{ne}$ in Eqs. (17) and (18) were estimated from linear regression as the best fit.

The errors of the estimates in Fig. (15) are considerable, such that it is difficult to identify the scaling of $d\mathcal{U}^{-2}/dt$. However, $d\mathcal{U}^{-2}/dt$ is clearly a decreasing function of the turbulence Reynolds number $Re$, as predicted by both equilibrium and non-equilibrium relations.

As far as $d\mathcal{L}^2/dt$ is concerned, points follow the non-equilibrium scaling. As the Reynolds number decreases both equilib-

460 rium and non-equilibrium predictions for $d\mathcal{L}^2/dt$ become close, so we expect that at even lower $Re$, the data will follow the

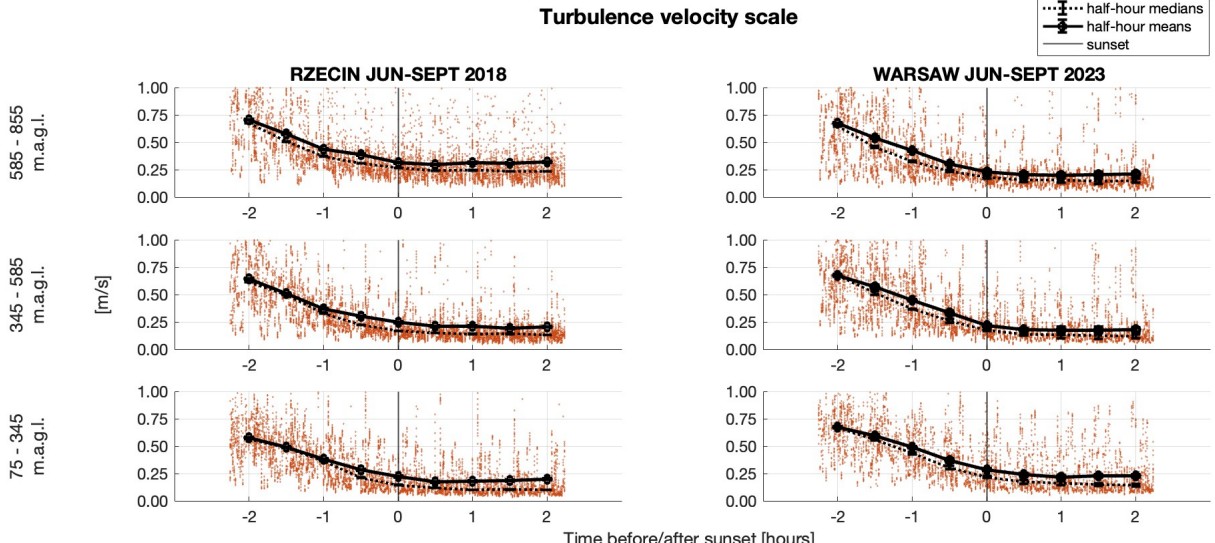

**Figure 13.** Standard deviation of vertical velocity component at different altitudes for data, mean, medians and error estimates (see Appendix A), measured during summer season (June-September) in, respectively, rural and urban environment at Rzecin PolWET station on 2018 (left column) and Warsaw Observatory Station on 2023 (right column).

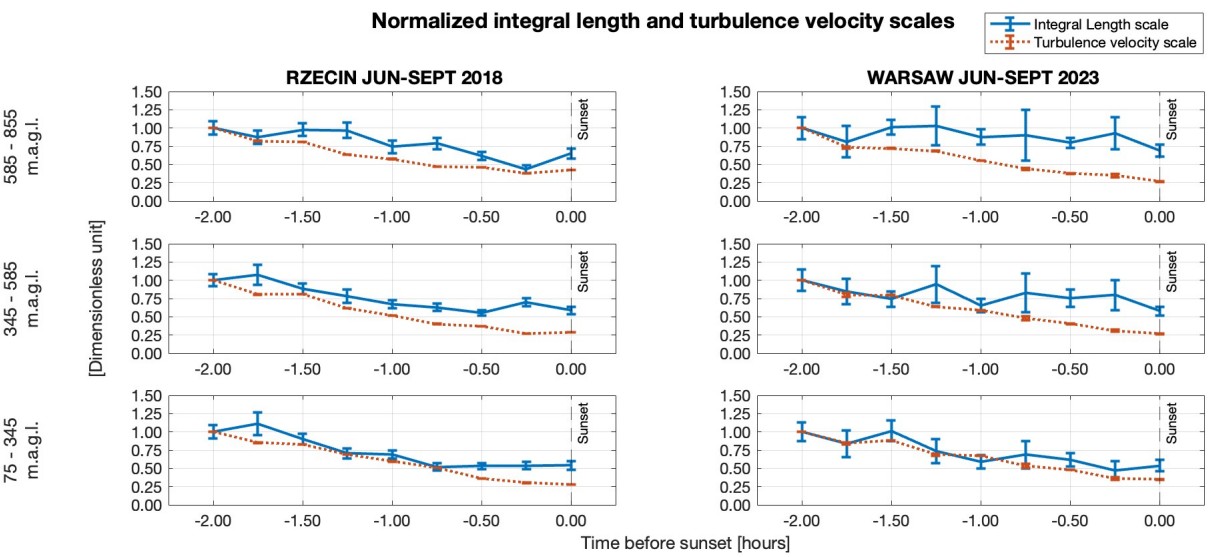

**Figure 14.** Integral length scales and turbulence velocity scales before the sunset normalised by the values measured at $t = -2$h, mean, medians and error estimates (see Appendix A). Rzecin PolWET station, rural environment (left column), Warsaw Observatory Station, urban environment (right column).

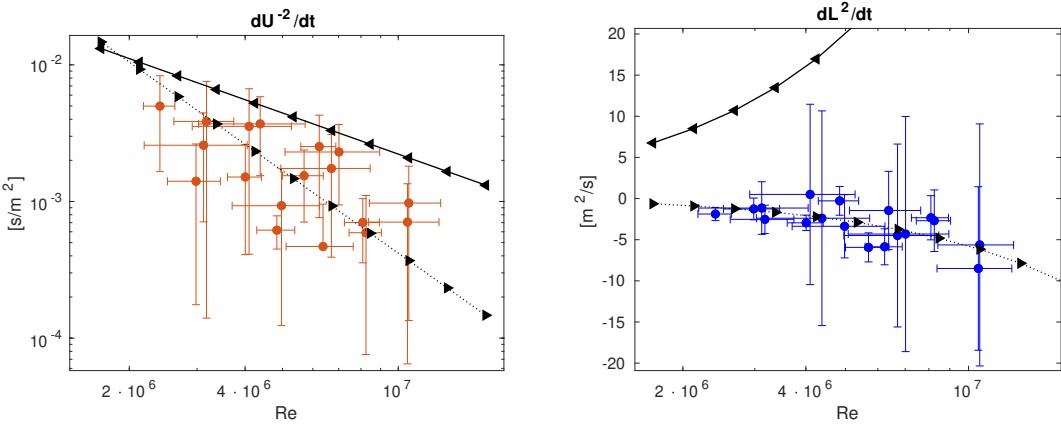

**Figure 15.** Left panel: Dependence of $\mathrm{d}\mathcal{U}^{-2}/\mathrm{d}t$ on the Reynolds number. Right panel: Dependence of $\mathrm{d}\mathcal{L}^2/\mathrm{d}t$ on the Reynolds number. Derivatives estimated from half-hour medians and error estimates (see Appendix A) (symbols with errorbars) compared to equilibrium relations (11) and (15) (solid lines) and non-equilibrium relations (17) and (18) (dashed lines).

equilibrium predictions. Hence, by analysis of both panels in Fig. (15), it can be concluded that the non-equilibrium decay is likely to be present in the initial stages.

### 5.5 Dissipation rates: comparison

Due to large deviations from the equilibrium scaling the energy dissipation estimates with the use of Eqs. (3) or (4) are not reliable at $t > -2\mathrm{h}$. Therefore we used Eq. (4) to calculate the profile of $\epsilon$ only at $t = -2\mathrm{h}$ and for $z/D < 0.6$. An example of the second-order structure function is presented in Fig. 16a and the dissipation rate, non-dimentionalized with the use of $w_*$ and the CBL height $D$ is presented in Fig. 16b, as a function of $z/D$. We used these estimates to compare with the predictions of Eqs. (1) and (2) in Figs. 17 and marked them as red dots at $t = -2\mathrm{h}$. Fig. 17 presents median dissipation values before the sunset obtained using the classical Taylor's law (Eq. 1) with $C_\epsilon = 0.5$ and assuming the non-equilibrium scenario. In the latter, dissipation rate is expected to follow the non-equilibrium relation (Eq. 2), until its values become equal to the predictions of the classical Taylor's law (Eq. 1), but with a higher value of the constant $C_\epsilon = 1$, which is the upper bound of the dissipation coefficient (Bos and Rubinstein, 2018). The dissipation rate further follows the equilibrium Taylor law (1) with $C_\epsilon = 1$. During this latter period, the frequency spectra will still deviate from the Kolmogorov's scaling at low wavenumbers (Steiros, 2022b). The same is true for the structure functions, cf. Eq. (7). The equilibrium $-5/3$ and $2/3$ slopes of spectra and structure functions will be reached asymptotically, at high wavenumbers which are not measured by the lidar system. By comparing both scaling laws, it can be concluded that the equilibrium Taylor law underpredicts the dissipation rate of turbulence kinetic energy up to a factor of 2.

The dissipation values decrease with altitude and in time. In Rzecin, the decrease is most rapid closer to the surface. Under the conditions of radiative cooling over a rural area, the surface-based inversion is developed, which stops the air-mixing

above the ground faster than in the urban sites. As an consequence of very similar meteorological conditions for both stations (dry and warm conditions), we assume that the differences in turbulence properties result mainly from differences in surface morphology between both of environments. Due to the difference in surface morphology, at the urban site the effect of wind shear and the presence of friction-driven turbulences are still significant at the altitudes 75-345 m.a.g.l. On the other hand, Fig. 17 suggests that at higher altitudes the dissipation rates decrease in time much faster over the urban site. Possibly, shortly 485 before the sunset the heat island effects are much more limited in height.

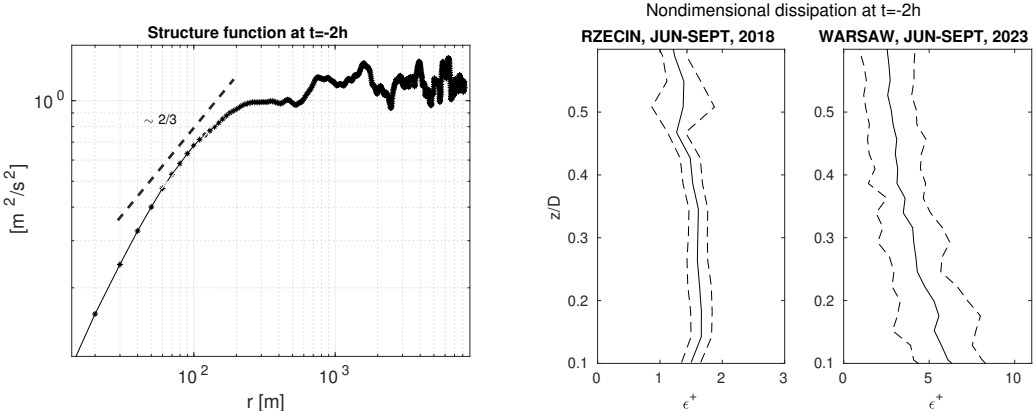

**Figure 16.** Exemplary second-order structure function at $t = -2$h and the nondimensionalized turbulence kinetic energy dissipation rates $\epsilon^+ = \epsilon D / w_*^3$ at $t = -2$h estimated from formula (4) (solid line) with uncertainty ranges (dashed lines) – see Appendix A.

## 6    Conclusions

The analysis performed in this work showed time, altitude, and surface-type dependencies of the properties of turbulences during the evening transition at the time span $\pm 2$h relative to the sunset. The calculated slopes of the frequency spectra and structure function of vertical wind before sunset deviate increasingly from the Kolmogorov's predictions, which agrees with 490 the non-equilibrium scenario. These values also deviate more with an altitude, implying the possible maximum height of the turbulence presence in the ABL, during its rapid decay. We argued that it is possible to explain the increasing deviations of the slopes with the use of the recent theories of turbulence. The crucial part was the observed decrease of the turbulence length scale, $\mathcal{L}$ during turbulence decay, which was also predicted by the non-equilibrium relations.

In this work we assumed that within the residual layer, where turbulence decays the production and turbulent transport of 495 kinetic energy are negligible, such that to the leading order the time change of kinetic energy is balanced by the dissipation rate. Under those assumptions the rates of change $\mathrm{d}\mathcal{U}^{-2}/\mathrm{d}t$ and $\mathrm{d}\mathcal{L}^{-2}/\mathrm{d}t$ can be expressed as functions of the turbulence Reynolds number $Re$. We choose as the initial condition $t = -2$h, when the estimated convective Deardorff scale $w_*$ is still larger than the friction velocity $u_*$. For such choice, our results suggest that statistics follow non-equilibrium relations before the sunset,

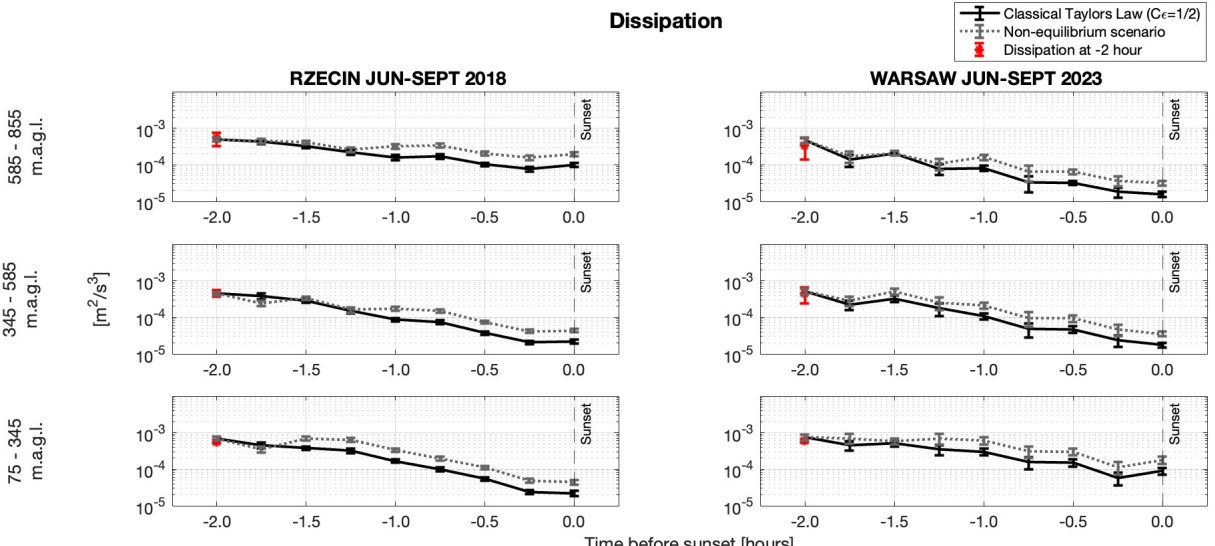

**Figure 17.** Turbulence kinetic energy dissipation rate calculated from the equilibrium Taylor law (1) and with the non-equilibrium scenario, with error estimates (see Appendix A) at different altitudes for data measured in, respectively, rural and urban environment at Rzecin PolWET station (left column) and Warsaw Observatory Station (right column). Dots at $t = -2$h are estimates from the second-order structure function.

when the turbulence Reynolds numbers are very high. Hence, non-equilibrium relations should be taken into account for the
estimation of the dissipation rate of turbulence kinetic energy in the initial stages of decay.

    Our work also shows differences in the turbulence properties between two different environments, the rural and urban. The latter is much more morphologically diverse and has a higher heat capacity. We found that over the urban area, turbulence is initially present at higher altitudes. Moreover, the convective time scale $\tau$ calculated 2 hours before the sunset is larger in Warsaw, which suggests that the decay of turbulence in the urban layer is slower, as compared to the rural one. However, our
observations suggest that it is the case only at lower altitudes. At heights 585–855 m.a.g.l. turbulence seem to decrease very rapidly over the urban site. Turbulence production by shear, which is affected by the surface morphology could also contribute to the difference between the two environments, especially at lower altitudes. We conclude that the influence of the surface morphology and heat capacity on turbulence decay is significant and should be accounted for in the parametrization schemes, which is in line with the results reported by Couvreux et al. (2016).

*Data availability.* The Doppler Lidar data from Rzecin PolWET site used in this study are published in the Zenodo repository:
HTTPS://DOI.ORG/10.5281/ZENODO.8181344 (Ortiz-Amezcua, P. and Alados-Arboledas, L., 2023)
under the Creative Commons Attribution 4.0 International license.
The Doppler lidar data from Warsaw Observatory Station are generated by the Aerosol, Clouds and Trace Gases Research Infrastructure

## Appendix A: Error estimates

In the analyses we used the standard error of the mean (SEM) of a quantity $\langle \phi \rangle$, defined as

$$\Delta \langle \phi \rangle_{sem} = \frac{\sigma}{\sqrt{N}}, \tag{A1}$$

where $\sigma$ is the standard deviation and $N$ is the number of samples. We assumed that the relative errors of median values are equal to the relative errors of the mean values. The uncertainties presented in Figs. 4 – 17 were determined as follows

- Errors of the ABL and CBL heights $D$ presented in Fig. 3 were estimated as SEM.

- Errors of the buoyancy fluxes from Fig. 4 were determined from a sum of relative SEM errors of temperature and temperature fluxes

$$\frac{\Delta \langle w'b' \rangle}{\langle w'b' \rangle} = \frac{\Delta \langle \theta_v \rangle_{sem}}{\langle \theta_v \rangle} + \frac{\Delta \langle w'\theta_v' \rangle_{sem}}{\langle w'\theta_v' \rangle}. \tag{A2}$$

- Error of the friction velocity $u_*$ in Fig. 5 was estimated as SEM, errors of the Deardorff velocity $w_*$ and the time scale $\tau$ are defined as

$$\frac{\Delta w_*}{w_*} = \frac{1}{3} \left( \frac{\Delta \langle w'b' \rangle}{\langle w'b' \rangle} + \frac{\Delta D_{sem}}{D} \right), \qquad \frac{\Delta \tau}{\tau} = \frac{\Delta w_*}{w_*} + \frac{\Delta D_{sem}}{D}. \tag{A3}$$

- To estimate errors of the turbulence velocity and length scales in Figs. 8, 12, 13 and 14 we took into account errors of vertical velocity measurements $\delta w$ obtained from the HALOlidar toolbox (Manninen, 2019) and errors of the mean. The half-hour averages $\mathcal{U}_{1/2}^2$ were calculated by averaging over time, and their measurement error $\delta \mathcal{U}_{1/2}^2$ is determined as

$$\mathcal{U}_{1/2}^2 \pm \delta \mathcal{U}_{1/2}^2 = \frac{1}{\Delta t} \int_t^{t+\Delta t} [w(t) + \delta w(t)]^2 \, \mathrm{d}t \approx \mathcal{U}_{1/2}^2 \pm \frac{1}{\Delta t} \int_t^{t+\Delta t} 2w(t)\delta w(t) \, \mathrm{d}t,$$

where $\Delta t = 1/2$h. Next, the mean values were calculated by averaging over $N$ half-hour averages. Error of the mean value was calculated from the sum of squares of the measurement error $\delta \mathcal{U}^2$ and the convergence error $\Delta \mathcal{U}_{sem}^2$

$$\Delta \mathcal{U}^2 = \left[ \left( \delta \mathcal{U}^2 \right)^2 + \left( \Delta \mathcal{U}_{sem}^2 \right)^2 \right]^{1/2}, \qquad \delta \mathcal{U}^2 = \frac{1}{N} \sum_{i=1}^N \delta \mathcal{U}_{1/2}^{(i)2}. \qquad \text{Further,} \qquad \frac{\Delta \mathcal{U}}{\mathcal{U}} = \frac{1}{2} \frac{\Delta \mathcal{U}^2}{\mathcal{U}^2}. \tag{A4}$$

The length scales $\mathcal{L}$ were also calculated based on the vertical velocity measurements and we assumed that their error consists of the measurement error $\delta\mathcal{L}$ and the convergence error $\Delta\mathcal{L}_{sem}$

$$\Delta\mathcal{L} = \left[(\delta\mathcal{L})^2 + (\Delta\mathcal{L}_{sem})^2\right]^{1/2}, \qquad \text{where} \qquad \frac{\delta\mathcal{L}}{\mathcal{L}} = \frac{\delta\mathcal{U}^2}{\mathcal{U}^2}. \tag{A5}$$

- Errors of the slopes in Figs. 9 and 10 and errors of the horizontal velocity in Fig. 11 where calculated as SEM.

- Derivatives presented in Fig. 15 were estimated as forward finite differences and their errors were estimated as

$$\Delta\frac{\mathrm{d}\mathcal{L}^2}{\mathrm{d}t} = 2\frac{\mathcal{L}(t+\Delta t)\Delta\mathcal{L}(t+\Delta t)}{\Delta t} + 2\frac{\mathcal{L}(t)\Delta\mathcal{L}(t)}{\Delta t}, \quad \Delta\frac{\mathrm{d}\mathcal{U}^{-2}}{\mathrm{d}t} = 2\frac{\mathcal{U}^{-3}(t+\Delta t)\Delta\mathcal{U}(t+\Delta t)}{\Delta t} + 2\frac{\mathcal{U}^{-3}(t)\Delta\mathcal{U}(t)}{\Delta t}. \tag{A6}$$

- Errors of the dissipation rate estimates in Fig. 16 are SEM.

- Errors of the dissipation rates presented in Fig. 17 read

$$\frac{\Delta\epsilon}{\epsilon} = 3\frac{\Delta\mathcal{U}}{\mathcal{U}} + \frac{\Delta\mathcal{L}}{\mathcal{L}}, \qquad \text{or} \qquad \frac{\Delta\epsilon}{\epsilon} = 2\frac{\Delta\mathcal{U}}{\mathcal{U}} + 2\frac{\Delta\mathcal{L}}{\mathcal{L}} \tag{A7}$$

in the equilibrium and non-equilibrium cases, respectively.

*Author contributions.* POA, MK, ŁJ, ISS and PP performed the measurements and collection of the data. MW, POA, and ISS worked on the concept of this work and the methodology. POA, MW, MK, and ISS contributed to the development of the code. POA and ŁJ developed the code and performed initial data analysis. MW contributed to the theory and its application in the data analysis and wrote the first draft of the manuscript. MK performed data curation and analysis. ISS, MW, and MK contributed to the interpretation of the results and prepared the figures. CBK and PP performed calculations of the fluxes, MW, MK, CBK and PP worked on the revision of the manuscript and answers of 550 Referees' comments. ISS instructed the measurements, supported the analysis, and provided effective and constructive comments to improve the manuscript. All authors contributed to the writing and revision of the text.

*Competing interests.* The authors declare no conflict of the interests.

*Acknowledgements.* PP, ISS and POA performed the measurements in Rzecin in 2018 within the Technical assistance for Polish Radar and Lidar Mobile Observation System (POLIMOS) funded by ESA-ESTEC Contract no. 4000119961/16/NL/FF/mg. We acknowledge the 555 provision of technical support of the PolWET site of PULS led by PI Bogdan Chojnicki. We thank for provision of the Doppler lidar system from GFAT-UGR group led by PI Lucas Alados-Arboledas.

POA, MK and ŁJ performed the measurements in Warsaw in 2023 with support of the European Commission under the Horizon 2020 – Research and Innovation Framework Programme with the ACTRIS-IMP project (G.A. no 871115) and ATMO-ACCESS (G.A. no. No 101008004).

POA, MK, MW, and ISS acknowledge the financial support of the National Science Centre, Poland through project 2021/40/C/ST10/00023 of programme SONATINA 5, the algorithm to obtain the turbulent properties from the Doppler Lidar measurements was developed within this project.

MW acknowledges financial support of the National Science Centre, Poland (Project No. 2020/37/B/ST10/03695) of programme OPUS 19, the theory and its application in data analysis were done within this project.

CBK and ISS acknowledge financial support of the National Science Centre, Poland under the Weave-UNISONO call in the Weave programme (Project No. 2021/03/Y/ST10/00206), the calculations of the surface flux in Warsaw were done within this project.

Experimental sites in Rzecin and Warsaw are a part of the Aerosol, Cloud and Trace Gases Research Infrastructure ACTRIS-ERIC (https://www.actris.eu/, last access 06/09/2024)

This research was funded in whole or in part by National Science Centre, Poland under the Weave-UNISONO call in the Weave pro-
gramme, 2021/03/Y/ST10/00206. For the purpose of Open Access, the authors have applied a CC-BY public copyright licence to any Author Accepted Manuscript (AAM) version arising from this submission.

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
