# Peer review of "Investigation of non-equilibrium turbulence decay in the atmospheric boundary layer using Doppler lidar measurements"

_EGUsphere, 2024_

## Referee Comment (RC1)

Karasewicz et al. have performed an analysis of Doppler lidar measurements of turbulence in the ABL just before and after sunset. They find that frequency spectra of vertical component of the wind deviates from -5/3 law for the inertial subrange, but is consistent with theories of non-equilibrium turbulence. They conclude that lassical models of turbulence decay tend to underestimate the dissipation rate of TKE in initial stages of decay. The study makes use of frequency spectra of vertical fluctuations and structure functions in rural and surface roughness- and heat island- affected urban environments before and after sunset.

Overall, I like this study. The authors have collected a very valuable Doppler lidar data set on turbulence in the ABL around sunset. While much effort has been expended in studying the ABL's evolution from morning till afternoon, there is a dearth of studies focusing on the changes in the ABL around sunset, when convective turbulence transitions to nocturnal turbulence, resulting in a shear-driven ABL near the ground and a residual layer elsewhere. As such, this study is an important addition to the literature.

That said, the manuscript and its impact could be improved significantly if the authors provide a proper context to the interpretation of the results and drawing of the necessary conclusions, by presenting the underlying velocity and length scales. What I mean is that the authors should interpret the results in the context of the prevailing convective velocity scale $w^*$, the friction velocity $u^*$ and the depth of the ABL $D$, as well as the height of roughness elements (especially in the urban environment). They should also provide error statistics (confidence intervals/error bars on plotted data points for better interpretation of the underlying secular trends. Detailed comments and suggestions can be found below.

I recommend that the manuscript be sent back for major revisions before acceptance for publication.

**Detailed Comments and Suggestions:**

Eqs. (9), (13) and (14): Kinematic viscosity coefficient $\nu$ is artificially introduced to bring in the Reynolds number $Re$. In reality $\nu$ must not appear in either equation, since high Reynolds number (asymptotic) turbulence, molecular viscosity should not appear explicitly except at Kolmogorov viscous scales. Now, if $Re$ is identified instead as TURBULENCE Reynolds number, then it is fine. This is because $Re$ is usually associated with mean flow velocity and the size of the turbulent layer.

How do we know $Re$ decreases during decay? Certainly $U$ decreases with time, but $L$ can increase or decrease, so that $U*L$ may not necessarily decrease. A better justification would be useful.

The central idea of the paper depends on integral scale $L$ decreasing with time under non-equilibrium turbulence decay. This must be proven beyond any doubt.

Eqs. (8) and (14) are valid only for isotropic turbulence since the implicit assumption is $u^2 = q^2/3$, where $q^2$ is twice the TKE.

L189-190: I don't understand this. In both Eqs.(13) and (15), time derivative of $L^2$ is NOT inversely proportional to $Re$. The statement is erroneous?

L255: While heat flux analysis is not central to this study, it is important in order to determine the Deardorff convective velocity scale $w*$, since the variance of vertical component of turbulence

velocity scales like $(w^*)^2$ under convection (and like friction velocity $u^*$ under shear). It is important to know what $w^*$ and the ABL depth $D$ are during the spectrum and structure function measurements. The paper should present their variation with time during the study period.

Figure 3: What is the height of the ABL? Important to know where exactly in the ABL, measurements are being made, Interior, near the top, near the surface layer, inside the surface layer?

L297 and Figure 4: "steeper than Kolmogorov?" Why? Instrumental errors? You mean "red_part of orange and yellow-part of green." Right?

Figure 4: How exactly do you fit the line to compute the slope? I can think of large systematic errors resulting, depending, for example, on the range of frequencies chosen to fit the spectrum in Figure 3.

Figure 4: What is the ABL height as a function of time? That would be useful in interpreting the results. For Warsaw, what is the height of roughness elements (buildings) above the ground? In other words, how close or far away are the measurement points from the ground and from the ABL top?

Figure 4: Extensive regions of steeper slopes during the actively heated, vigorous ABL. Why?

Figure 5: Would be useful to plot also in a third panel, the product of turbulence Velocity scale and integral length scale, since this is proportional to the "Reynolds number $Re$". Better call it turbulence Reynolds number to distinguish it from conventional Reynolds number based on mean velocity. This is important in view of the statements made earlier on how Reynolds number behaves during decay.

Figure 5: Error bars on the black dots (median values) are essential to interpret the variability with time of both scales, especially the length scale. The scatter in the length scale is much larger than in the velocity scale and so the trend indicated is not so easy to interpret in the absence of error bars.

L315-316: This interpretation is not so obvious, in the absence of confidence intervals (error bars).

L320: The cited heights must be put in proper context by specifying the average height of roughness elements and the ABL height.

Figures 5 – 7: Why median values instead of mean values? Perhaps both? Please justify.

Figures 5 – 7: Information on ABL height and roughness elements would be useful in digesting the results.

Figures 5-7: Just before sunset, the buoyancy flux decreases rapidly, because of how the solar radiation decreases toward the end of the heating cycle. This itself can lead to non-equilibrium conditions, even before the heating is cut off at sunset ($t = 0$). This should be mentioned.

Figures 5 – 7: The characteristic time scale in the ABL is the eddy turnover time scale, proportional to $D/w^*$, where $D$ is the depth of ABL and $w^*$ is the convective (Deardorff) velocity scale. It is useful to know what this time scale is, since with cutoff of heating, one can expect

convective turbulence to decay and lead to the formation of a residual layer with shear-driven layer adjacent to the surface, on this time scale.

Figures 5 – 7 and L 327-339: Clearly, the departure from -5/3 (Figure 6) appears to increase with height at Rzecin, whereas Warsaw does not show this tendency. Information on ABL height and height of roughness elements would be useful to understand this.

Figure 8 and L345-350: The mean horizontal velocity appears to be roughly 5 m/s, remarkably uniform with height and measurement location. Anyway, the corresponding friction velocity $u^*$ is likely to be 0.15 – 0.25 m/s (say 0.2 m/s on the average). What are the values of $w^*$ during this time? This is important to know since the ratio $w^*/u^*$ must be high enough to assume convective nature of turbulence. Otherwise, $u^*$ complicates interpretation of the results.

L366: What is the value of $H$?

Figure 10: Plot of $w^*$ and $u^*$ with time relative to sunset would be useful here, since $w^*$ is likely more relevant before sunset and $u^*$ after. In other words, the underlying scales would be useful in better understanding of the results.

Figure 11: Why choose $U_0$ and $L_0$ at -2 hr? The time chosen should be justified in comparison with the eddy turnover time scale $D/w^*$. Without error bars, it is hard to interpret the secular trend, especially of integral length scale.

Figure 12: This is the "piece de resistance" of this study. While the conclusions to be drawn from these plots are clear, it would help to remember the assumptions and data processing and interpretations that preceded this. Also, the data points should have at least approximate confidence limits (error bars) for completeness.

L378: I don't believe Reynolds number is a relevant quantity in asymptotic turbulence that exists at high enough Reynolds numbers. In Figure 12, Re is just $U*L$ normalized by a constant value of kinematic viscosity anyway and so it is indicative of $U*L$, the quantity proportional to turbulent viscosity. Therefore Re is more appropriately turbulence Reynolds number.

Figure 13: These results follow naturally from results in Figure 12 and presumed behavior according to Eqs. (1) and (2). However, the choice of $U_0$ and $L_0$ are crucial to the interpretation. Some justification is needed as to their choice at 2 hrs before sunset (see above) in terms of the eddy turnover time scale.

Conclusions: It appears to me that the conclusions must be tempered a bit and justified better. What I mean is this:

1. One can expect a drastic change of turbulence characteristics in the ABL (transition from convective turbulence in the entire ABL to residual layer and shear-driven turbulence near the surface) immediately after sunset, leading to non-equilibrium conditions. This transition is not adequately addressed, since the governing velocity scale changes from $w^*$ to $u^*$ near the surface, and turbulence (non-intermittent) presumably dies down in the residual layer. As such, some discussion of $u^*$ is essential, instead of lumping everything into a "Reynolds number" based interpretation.
2. The results suggest non-equilibrium conditions are present BEFORE sunset, since data at 2 hr before sunset is used as initial conditions. However, the behavior of turbulence between -2 hr and 0 hr depends critically on how $w^*$ decreases during this time. It is

likely that $w^*$ decreases more rapidly during this period than during mid-morning to mid-afternoon conditions. In any case, without knowing $w^*$ and $D$, it is hard to interpret the results and unconditionally agree with the conclusions.

Minor Points:

L1 & L12:  Replace "short" by "shortly"
L20: Typo - should be homogeneous.
L109: Inertial subrange is a more appropriate term.

Eq. (14): $\dfrac{\nu}{UL}$ must be $\left(\dfrac{\nu}{UL}\right)^2$

---

## Author Comment (AC1)

**Response to Prof. Lakshmi Kantha**

We would like to thank Prof. Lakshmi Kantha for the comments and suggestions to our manuscript. We addressed all the comments and performed additional data analyses and calculations as suggested by the Referee. Please note that the comments by the Referee are presented in blue color and with the italic font, whereas our responses are written in black and with the sans-serif font.

1. *Overall, I like this study. The authors have collected a very valuable Doppler lidar data set on turbulence in the ABL around sunset. While much effort has been expended in studying the ABL's evolution from morning till afternoon, there is a dearth of studies focusing on the changes in the ABL around sunset, when convective turbulence transitions to nocturnal turbulence, resulting in a shear-driven ABL near the ground and a residual layer elsewhere. As such, this study is an important addition to the literature.*

   We would like to thank the Referee for the positive overall judgment of the manuscript.

2. *That said, the manuscript and its impact could be improved significantly if the authors provide a proper context to the interpretation of the results and drawing of the necessary conclusions, by presenting the underlying velocity and length scales. What I mean is that the authors should interpret the results in the context of the prevailing convective velocity scale w\* , the friction velocity u\* and the depth of the ABL D, as well as the height of roughness elements (especially in the urban environment). They should also provide error statistics (confidence intervals/error bars on plotted data points for better interpretation of the underlying secular trends.*

   As suggested by the Referee, we strived to improve the manuscript by presenting the underlying convective velocity and the friction velocity scales, the depth of the ABL and the height of the roughness elements. In the revised version we added new Section 5.1 with the new results, we also provided error statistics.

3. *Eqs. (9), (13) and (14): Kinematic viscosity coefficient $\nu$ is artificially introduced to bring in the Reynolds number Re. In reality $\nu$ must not appear in either equation, since high Reynolds number (asymptotic) turbulence, molecular viscosity should not appear explicitly except at Kolmogorov viscous scales. Now, if Re is identified instead as TURBULENCE Reynolds number, then it is fine. This is because Re is usually associated with mean flow velocity and the size of the turbulent layer.*

   The viscosity coefficient $\nu$ was introduced in the equations to bring the Reynolds number Re and to non-dimensionalize the equations. In this procedure, we followed the previous work [2] on turbulence theory. In our study the

Reynolds number Re is defined based on the turbulence velocity scale (standard deviation of the vertical velocity component) and the turbulence integral length scale. In the revised version we explicitly wrote in line 163 that $Re$ is the turbulence Reynolds number, which is proportional to the ratio of the eddy and molecular viscosities (it is not associated with mean flow velocity and the size of the turbulent layer).

4. *How do we know Re decreases during decay? Certainly U decreases with time, but L can increase or decrease, so that U\*L may not necessarily decrease. A better justification would be useful.*

We expect that the product U\*L will decrease in time, in spite of the fact that L may increase in time. This can be justified as follows. In case of the equilibrium decay of isotropic turbulence the change of the length scale and the turbulence velocity scales are described by Eqs. (11) and (15) in the manuscript. After rearranging and combining the two equations we obtain the following formula for the change of the product U\*L in time:

$$\mathcal{U}\frac{\mathrm{d}\mathcal{L}}{\mathrm{d}t} + \mathcal{L}\frac{\mathrm{d}\mathcal{U}}{\mathrm{d}t} = \frac{\mathrm{d}\mathcal{U}\mathcal{L}}{\mathrm{d}t} = -\frac{1}{3}C_\epsilon\mathcal{U}^2 + \frac{A_e}{2}\mathcal{U}^2$$

Using the definition of the coefficient $A_e$ given in line 177 of the manuscript we finally obtain

$$\frac{\mathrm{d}\mathcal{U}\mathcal{L}}{\mathrm{d}t} = -\frac{1}{15}C_\epsilon\mathcal{U}^2 < 0.$$

Hence, in the case of the equilibrium decay the above formula predicts that the product U\*L decreases in time. In the initial stages of the non-equilibrium decay both U and L decrease in time and so does the product U\*L. As time progresses L will increase in time, however according to Ref. [2] the rate of increase is smaller or equal to the corresponding equilibrium rate. In the revised manuscript we added explanations in lines 179-180 and added formula (16).

5. *The central idea of the paper depends on integral scale L decreasing with time under non- equilibrium turbulence decay. This must be proven beyond any doubt.*

As shown in Fig. 8 in the manuscript, the calculated integral length scales decrease with time independent of the chosen averaging window. In order to prove it, in the revised manuscript we added error bars on the plots.

6. *Eqs. (8) and (14) are valid only for isotropic turbulence since the implicit assumption is $u^2 = q^2/3$ , where $q^2$ is twice the TKE.*

Indeed, these equations are valid for the isotropic turbulence. It is usually assumed that turbulence in the atmospheric boundary layer is locally isotropic at small scales (possibly, up to the order of tens of meters). However, with only one, vertical component of velocity fluctuations measured with the frequency of 1 Hz, we cannot verify this assumption or investigate how the anisotropy

affects the results. However, even if the anisotropy is present initially due to the mechanisms of turbulence production, turbulence will return to isotropy during its decay, after the production mechanisms are released [3, 4]. Hence, we argue that the assumptions used in this study, although simplified, still reflect the general trends observed in the experimental results.

7. *L189-190: I don't understand this. In both Eqs.(13) and (15), time derivative of L2 is NOT inversely proportional to Re. The statement is erroneous?*

   The statement was erroneous. In the revised version we wrote in lines..... that the time derivative of $\mathcal{L}^2$ "...is a decreasing funcion of $Re$. "

8. *L255: While heat flux analysis is not central to this study, it is important in order to determine the Deardorff convective velocity scale w\*, since the variance of vertical component of turbulence velocity scales like $(w*)^2$ under convection (and like friction velocity u\* under shear). It is important to know what w\* and the ABL depth D are during the spectrum and structure function measurements. The paper should present their variation with time during the study period.*

   As requested by the Reviewer, we calculated the convective velocity scale $w_*$ and the friction velocity $u_*$ from independent measurements of velocity, temperature and the mixing ratio in the surface layer, both in the Warsaw and Recin sites. In the revised manuscript we presented their variation in time, relative to the sunset. New results are presented Section 5.1 the revised manuscript.

9. *Figure 3: What is the height of the ABL? Important to know where exactly in the ABL, measurements are being made, Interior, near the top, near the surface layer, inside the surface layer?*

   Fig. 3 presents the exemplary frequency spectra of vertical wind measured on 28.06.2018 at Rzecin PolWET station at the height of 195 m.a.g.l. at 18:30 and 19:30. The height of ABL estimated for this particular time equals approximately 630–700m. The surface layer corresponds to the lowest part of the ABL (below 100m), hence the altitude 195 m.a.g.l. where the measurements were made corresponds to the interior of the ABL. We added this information in the revised manuscript, in line 352.

10. *L297 and Figure 4: "steeper than Kolmogorov?" Why? Instrumental errors? You mean "red part of orange and yellow-part of green." Right?*

    In our opinion, the frequency spectra are steeper than Kolmogorov's prediction mainly because of the effective low-pass filtering connected with the finite resolution in time and space, see also [1] (reference cited in the manuscript). We presume that the steepening is not caused by any physical mechanisms, although further studies would be needed to investigate this issue. In the revised version we corrected description of the colormap in Fig. 7.

11. *Figure 4: How exactly do you fit the line to compute the slope? I can think of large systematic errors resulting, depending, for example, on the range of frequencies chosen to fit the spectrum in Figure 3.*

To compute the slope we fit the line in the following range of frequencies $f \in [0.15, 0.3]$. Our criteria were to calculate the slope of the small-scale part of the spectrum, were Kolmogorov scaling sholud be observed, but at the same time, to avoid the part of the spectrum which is affected by the aliasing. A different chioce of the fitting range will affect the results, however, as we discussed in Section 2.4 we focused on detecting changes of the slopes during the dacey rather than on their exact values. In the revised version we added information about the fitting range in line 303.

12. *Figure 4: What is the ABL height as a function of time? That would be useful in interpreting the results. For Warsaw, what is the height of roughness elements (buildings) above the ground? In other words, how close or far away are the measurement points from the ground and from the ABL top?*

In the revised manuscript we added the estimates of the ABL height in figure 3 and added information about the height of roughness elements in lines 403-405. According to the information from the Copernicus Land Monitoring Services https://land.copernicus.eu/en/products/urban-atlas/building-height-2012#general_info the average height of the urban canopy for two grids in Warsaw, both centred in the RS Lab: one measuring 1 km x 1 km area and the other measuring the area of 2.5 km x 2.5 km, the estimated mean building height equals 12.16 m and 13.35 m, respectively. For the first grid the maxima and minima are approximately 82.2 m and 4.0 m and the standard deviation is 6.86 m. For the second grid the maxima and minima are approximately 219.0 m and 3.0 m and the standard deviation is 10.16 m. In Rzecin, there is a reed 2-2.5 m heigh to the north of the EC tower (4.5m heigh), and to the south of the tower there are low sedges max. 0.5 m high.

13. *Figure 4: Extensive regions of steeper slopes during the actively heated, vigorous ABL. Why?*

The steeper slopes of the frequency spectra follow probably from the finite spatial and temporal resolution of the measurement, see e.g. [1]. As we discussed in Section 5, the steep slopes of frequency spectra in the convective regime were also reported by other authors (Darbieu et al., 2015). There were also speculations about the possible presence of the steeper, $-11/5$ Bolgiano-Obukhov scaling in convective turbulence. However, with the available, relatively low spatio-temporal resolution of the data we are not able to answer definitely why the slopes are steeper than $-5/3$.

14. *Figure 5: Would be useful to plot also in a third panel, the product of turbulence Velocity scale and integral length scale, since this is proportional to the "Reynolds number Re". Better call it turbulence Reynolds number to distinguish it from conventional Reynolds number based on mean velocity.*

*This is important in view of the statements made earlier on how Reynolds number behaves during decay.*

We added the third panel the product of $\mathcal{U}$ and $\mathcal{L}$ in Fig. 8.

15. *Figure 5: Error bars on the black dots (median values) are essential to interpret the variability with time of both scales, especially the length scale. The scatter in the length scale is much larger than in the velocity scale and so the trend indicated is not so easy to interpret in the absence of error bars.*

We added error bars on the plots.

16. *L315-316: This interpretation is not so obvious, in the absence of confidence intervals (error bars).*

With the error bars in Fig. 8 we can now better justify the choice of the detrending window. To calculate time derivationes of L and U with a good accuracy the statistical errors should be reduced as much as possible. Still, the general tendencies visible in Fig. 8 are the same for all windows.

17. *L320: The cited heights must be put in proper context by specifying the average height of roughness elements and the ABL height.*

In order to provide the proper context, we reformulated the beginningof Section 5.4, lines 590–595.

18. *Figures 5 – 7: Why median values instead of mean values? Perhaps both? Please justify.*

Median values are advantageous if the probability distribution of data is non-Gaussian or in the presence of rare but very large or very small values (outliers), which affect the mean. In the revised manuscript we present both, the mean and median values in figures 8–13 and added a comment in line 375.

19. *Figures 5 – 7: Information on ABL height and roughness elements would be useful in digesting the results.*

We added this information.

20. *Figures 5-7: Just before sunset, the buoyancy flux decreases rapidly, because of how the solar radiation decreases toward the end of the heating cycle. This itself can lead to non-equilibrium conditions, even before the heating is cut off at sunset (t = 0). This should be mentioned.*

In the revised manuscript we calculated the buoyancy flux in Fig. 4 and showed that it becomes negative or close to zero for $t > -2$h, when the evening transition starts. In the revised version we comment on it in Section 5.1 and comment in lines 350 that in the absence of forcing the changes of U and L can be described by formulas derived in Sections 2.2 and 2.3.

21. *Figures 5 – 7: The characteristic time scale in the ABL is the eddy turnover time scale, proportional to D/w\*, where D is the depth of ABL and w\* is the convective (Deardorff) velocity scale. It is useful to know what this time scale is, since with cutoff of heating, one can expect convective turbulence to decay and lead to the formation of a residual layer with shear-driven layer adjacent to the surface, on this time scale.*

    The estimated eddy turnover time scale is presented in the new figure 5. It increases sharply around $t = -2$ to values $0.3$h in Rzecin and $0.6$ in Warsaw.

22. *Figures 5 – 7 and L 327-339: Clearly, the departure from -5/3 (Figure 6) appears to increase with height at Rzecin, whereas Warsaw does not show this tendency. Information on ABL height and height of roughness elements would be useful to understand this.*

    Our results show that the height of the mixed layer in Warsaw is larger than in Rzecin. Moreover, the convective velocity scale $w_*$ is also larger. This indicates that stronger, more vigorous turbulence is present in the urban ABL. This causes differences in the slopes. The slopes of frequency spectra in Rzecin site at larger altitudes may be more affected by the stable stratification above the ABL top.

23. *Figure 8 and L345-350: The mean horizontal velocity appears to be roughly 5 m/s, remarkably uniform with height and measurement location. Anyway, the corresponding friction velocity u\* is likely to be 0.15 – 0.25 m/s (say 0.2 m/s on the average). What are the values of w\* during this time? This is important to know since the ratio w\*/u\* must be high enough to assume convective nature of turbulence. Otherwise, u\* complicates interpretation of the results.*

    The estimated values of w* in Fig. 5. They decrease rapidly before the sunset. The friction velocity equals approximately 0.2 – 025 m/s. The Reviewer asks about the values of the w*/u* ratio, which should be high enough to assume the convective nature of turbulence. However, our study focuses on the time period before the sunset, where the buoyancy forcing is absent, therefore turbulence changes its nature from convective to decaying (or shear-forced in the surface layer). We estimated the ratio w*/u* and in fact it becomes smaller than unity at time at $t > -2h$.

24. *L366: What is the value of H?*

    We added information about $H$ in line 441.

25. *Figure 10: Plot of w\* and u\* with time relative to sunset would be useful here, since w\* is likely more relevant before sunset and u\* after. In other words, the underlying scales would be useful in better understanding of the results.?*

    We added the plot of w* and u* measured in the surface layer in Warsaw and in Rzecin. In the investigated period of time, we observe that w* decreases rapidly and becomes smaller than the friction velocity at time -1h − -0.5h.

This rapid decrease of forcing is of our interest in the present study. The dominant friction velocity u* indicates that turbulence is still produced by shear. However, we expect that this type of forcing is localized close to the surface and that at larger altitudes turbulence decays according to Eq. (10), i.e. the forcing terms are negligible in the budget of TKE.

26. *Figure 11: Why choose U0 and L0 at -2 hr? The time chosen should be justified in comparison with the eddy turnover time scale D/w\*. Without error bars, it is hard to interpret the secular trend, especially of integral length scale.*

    We justified the choice of the time -2h using the calculated fvalues of $w_*$ and $u_*$ in Scetion 5.1. Short after $t = -2h$ the buoyancy flux becomes negligible and the evening transition starts. Other choice of U0 an L0 would rescale the plots, however it should not affect the observed trends. In the revised version we added error bars to interpret the secular trend.

27. *Figure 12: This is the "piece de resistance" of this study. While the conclusions to be drawn from these plots are clear, it would help to remember the assumptions and data processing and interpretations that preceded this. Also, the data points should have at least approximate confidence limits (error bars) for completeness.*

    In the revised version we added the confidence limits on the plots and commented on the assumptions and data processing.

28. *L378: I don't believe Reynolds number is a relevant quantity in asymptotic turbulence that exists at high enough Reynolds numbers. In Figure 12, Re is just U\*L normalized by a constant value of kinematic viscosity anyway and so it is indicative of U\*L, the quantity proportional to turbulent viscosity. Therefore Re is more appropriately turbulence Reynolds number.*

    The Re number which we use in the present study is indeed proportional to the ratio of the turbulent and molecular viscosities. This parameter can be used to characterize turbulence. In the revised version we wrote in line 163 that $Re$ is the turbulence Reynolds number, which is proportional to the ratio of the eddy and molecular viscosities.

29. *Figure 13: These results follow naturally from results in Figure 12 and presumed behavior according to Eqs. (1) and (2). However, the choice of U0 and L0 are crucial to the interpretation. Some justification is needed as to their choice at 2 hrs before sunset (see above) in terms of the eddy turnover time scale.*

    We justified the choice of the time -2h using the calculated fvalues of $w_*$ and $u_*$ in Scetion 5.1.

30. *One can expect a drastic change of turbulence characteristics in the ABL (transition from convective turbulence in the entire ABL to residual layer*

*and shear-driven turbulence near the surface) immediately after sunset, leading to non-equilibrium conditions. This transition is not adequately addressed, since the governing velocity scale changes from w\* to u\* near the surface, and turbulence (non-intermittent) presumably dies down in the residual layer. As such, some discussion of u\* is essential, instead of lumping everything into a "Reynolds number" based interpretation.*

In the revised version we calculated both w\* and u\* in the surface layer and we show that the transition from the convective to residual, decaying turbulence or shear driven turbulence near the surface is present already before the sunset. We added a disussion in Section 5.1, lines 345-350.

31. *The results suggest non-equilibrium conditions are present BEFORE sunset, since data at 2 hr before sunset is used as initial conditions. However, the behavior of turbulence between -2 hr and 0 hr depends critically on how w\* decreases during this time. It is likely that w\* decreases more rapidly during this period than during mid-morning to mid- afternoon conditions. In any case, without knowing w\* and D, it is hard to interpret the results and unconditionally agree with the conclusions.*

We show that w\* is almost constant for $t = -6h--...-3$ and next decreases very rapidly, we also estimated the CBL height and u\*. In the revised veriosn we modified conclusions in this context, see lines 490-495.

32. *Minor Points: L1 & L12: Replace "short" by "shortly"*
    *L20: Typo - should be homogeneous.*
    *L109: Inertial subrange is a more appropriate term.*
    *Eq. (14): $\frac{\nu}{\mathcal{UL}}$ must be $\left(\frac{\nu}{\mathcal{UL}}\right)^2$*

We introduced all the corrections.

**References**

[1] Banakh, V.A., Smalikho, I.N., Falits, A.V. and Sherstobitov, A.M. 2021 Estimating the Parameters of Wind Turbulence from Spectra of Radial Velocity Measured by a Pulsed Doppler Lidar. Remote Sens. 13, 2071.

[2] Steiros, K., 2022b. Turbulence near initial conditions. Phys. Rev. Fluids. 10, 104607. https://link.aps.org/doi/10.1103/PhysRevFluids.7.104607

[3] Lumley, J.L., Newman, G.R., 1977 The return to isotropy of homogeneous turbulence. Journal of Fluid Mechanics. 82, 161–178. doi:10.1017/S0022112077000585

[4] Yang P.-F., Pumir, A., Xu, H., 2021 Return to isotropy of homogeneous shear-released turbulence. Phys. Rev. Fluids, 6, 044601, https://link.aps.org/doi/10.1103/PhysRevFluids.6.044601

---

## Author Comment (AC2)

**1 Response to Reviewer #2**

We would like to thank Reviewer #2 for the comments and suggestions to our manuscript. We addressed all the comments, provided additional explanations and analyses, as requested by the Referee. Please note that the comments by the Referees are presented in blue color and with the italic font, whereas our responses are written in black and with the sans-serif font.

1. *This is a well-written manuscript containing thought-provoking results. However, there are several fundamental issues which need to be addressed before the paper can be accepted.*

   We would like to thank the Referee for the positive overall judgement of the manuscript, we tried our best to address the fundamental issues mentioned by the Referee and improve the manuscript accordingly.

2. *The authors borrowed newly developed non-equilibrium theories from turbulence literature and applied them to boundary layer turbulence. In the original theoretical development, the buoyancy effects are not included. So, the authors neglected the effects of stratification altogether (see their comment on page 11). In a transitional boundary layer, the impacts of atmospheric stability cannot be neglected. In the revised manuscript, some efforts must be made to include the effects of stability in the derivations [e.g., Eq. (8)].*

   This comment addresses the buoyancy effects which were not included in the theoretical developments in the original manuscript. In the revised manuscript we included the buoyancy $B$, the turbulent transport $T$ and the shear forcing $P$ in the kinetic energy equation (8).
   We performed additional data analyses and we were able to estimate the buoyancy forcing $B$ from measurements which were performed in parallel, by sonic anemometers placed on a meteorological mast in Rzecin site and at the roof of the building of the Institute of Geophysics in Warsaw.
   The new results shown in new Figure 4 suggest that the buoyancy forcing was still positive at time $t = -2h$, i.e. two hours before the sunset, but short after it became negative in Rzecin and very close to zero in Warsaw. Nadeau et al. (2021) defined this moment as the beginning of the evening transition, where turbulence start to decay more rapidly than in the afternoon. Since our focus is on times $t > -2h$, which correspond to the evening transition, we neglected the buoyancy forcing in further theoretical analyses in Section 2.2. We were not able to estimate the turbulent complete transport term $T$, but we assumed that it also becomes small in the absence of forcing. The shear forcing $P$ is expected to play a role only relatively close to the Earth surface, hence both $T$ and $P$ were neglected and we assumed that to the leading order the decay of turbulence kinetic energy is described by Eq. (10). In the revised manuscript we added discussions in Section 2.2, lines 156-160.

3. *The authors briefly mentioned 3rd-order structure functions in the manuscript. I would like to see evidence that the Lidar data conform to the 4/5th law (Karman-Howarth equation) prior to evening transition.*

To calculate the 3rd-order structure functions and check if the Lidar data conform to the 4/5th law the fluctuations of the longitudinal velocity component (i.e. along the mean wind direction) are needed. The doppler Lidar system measures with the frequency of $1Hz$ only one, radial (i.e. along the beam) component of the wind velocity. The horizontal wind was estimated only every 30 minutes, during the Vertical Azimuth Display (VAD) scans with a constant elevation of 70∘ and based on 12 azimuth points. The low frequence of these measurements does not allow to estimate the fluctuations of the horizontal wind component and calculate the 3rd-order structure function.

4. *Give at least a few examples of EDR estimated via second-order and third-order structure functions and compare them against Equations (1) and (2). Please clearly show the structure functions and fitted slopes in the revised manuscript. [Add these materials in Section 5.3].*

As discussed in the reply to comment 3, we could not calculate the 3rd order structure functions, becaue only thevertical (transverse) velocity component was measured with a sufficient frequency. In the revised version we calculated EDR via the second order structure function at $t = -2h$, when the scaling was still relatively close to the Kolmogorov scaling and show the structure functions, fitted slopes and the profiles of EDR in the new figure 16. At $t > -2$h the slopes become smaller than $2/3$ and estimating EDR based on the Kolmogorov's equilibrium assumptions is not justified, in our opinion.

5. *Elaborate on the (relative) accuracy of longitudinal and vertical velocity estimation from Lidar observations. Give references.*

In the revised version we added information on the accuracy of velocity estimation from Lidar observations in lines 284-285, and added references to works by Rye and Hardesty (1993) and Pearson et al. (2009). In the revised manuscript we also calculated the standard errors of the mean of all the calculated variables and added error bars on plots.

6. *Do the results hold for the longitudinal velocity component? Why not?*

As explained in the reply to comment 3, only vertical velocity component was measured with the frequency of $1Hz$. The profiles of the horizontal component of the wind velocity were estimated only every 30 minutes.

7. *The empirical Equation (1) is conventionally used in conjunction with TKE. What is the justification for using it only for the vertical velocity scale? One cannot invoke isotropy here.*

The Reviewer addresses the assumption of isotropy. Indeed, atmospheric turbulence is isotropic only at relatively small scales. However, with only one, vertical component measured with the sufficient frequencey, we were only able to estimate the vertical velocity scale. We still hope that the derived formulas

can describe the scaling of statistics during the dacay of turbulence, however, the constants can be somewhat affected.

8. *Figure 4 (left panel): in the entire convective boundary layer, the inertial-range slope is close to -2. Why? Can we trust the observational data?*

The observed inertial-range slopes of the frequency spectra are indeed close to -2 in the convective boundary layer. In our opinion, the steepening of the spectra is due to instumental issues, in particular due to the finite spatio-temporal resolution of the measurements. The same was argued e.g. by Banakh et al. (2021). The averaging in space and time acts like a low-pass filter, which affects the measured part of the spectra. Additionally, the slopes of the structure functions and the frequency spectra can be affected in different ways. Moreover, the chosen range of scales were the slopes are calculated also affects results. Hence, we wrote in lines 230–232 that we focus on the changes of the scaling in time, rather than on the exact values of the slopes.